# BEDTIME: A UNIFIED BENCHMARK FOR AUTOMATICALLY DESCRIBING TIME SERIES

## ABSTRACT

Recent works propose complex multi-modal models that handle both time series and language, ultimately claiming high performance on complex tasks like time series reasoning and cross-modal question-answering. However, they skip evaluations of simple and important foundational tasks, which complex models should reliably master. They also lack direct, head-to-head comparisons with other popular approaches. So we ask a simple question: *Can recent models even produce generic visual descriptions of time series data?* In response, we propose three new tasks, posing that successful multi-modal models should be able to *recognize*, *differentiate*, and *generate* language descriptions of time series. We then create BEDTime, the first benchmark dataset to assess models on each task, comprising four datasets reformatted for these tasks across multiple modalities. Using BEDTime, we evaluate 13 state-of-the-art models, and find that (1) surprisingly, dedicated time series foundation models severely underperform, despite being designed for similar tasks, (2) vision–language models are quite capable, (3) language-only methods perform worst, despite many lauding their potential, and (4) all approaches are clearly fragile to a range of realistic robustness tests, indicating avenues for future work.[*]

## 1 INTRODUCTION

Time series measure how environments change as numerical data is collected over time. Decision-making in crucial domains, like medicine (Lu et al., 2023; Romanowski et al., 2023; Yapa et al., 2022; Trinh et al., 2022; Rackoll et al., 2021) and finance (Hu et al., 2020; Ślepaczuk & Zenkova, 2018; Ji et al., 2019; Sezer & Ozbayoglu, 2018; Schulmeister, 2019), hinge on accurate time series analysis. However, the history of machine learning models for time series, regardless of task, has mostly been uni-modal (Wang et al., 2024a; Tan et al., 2024; Jiang et al., 2025; Wang et al., 2024b). So many recent works have instead proposed multi-modal models that can "reason" about time series using language and/or solve new multi-modal tasks (Merrill et al., 2024; Chow et al., 2024; Jin et al., 2023; Yu et al., 2023; Wang et al., 2025; Xie et al., 2025; Tan et al., 2025; Xue & Salim, 2023). This approach opens doors to performing more-complex tasks, but makes evaluations extremely challenging and important.

Prior works on language-based time series models have two key limitations in how models are evaluated. First, most models are proposed alongside specialized evaluation datasets and evaluated in near-isolation, leaving a lack of head-to-head comparisons between models. Such comparisons are crucial to track progress. Second, dedicated benchmarking efforts (Merrill et al., 2024; Cai et al., 2024; Fons et al., 2024; Tan et al., 2025) often focus on complex reasoning tasks, but these can obscure the core abilities of the models. Evaluation of simpler, more fundamental, tasks are needed to better isolate and assess a model's true capabilities in time series understanding and reasoning.

We propose that describing key visual features in univariate time series data is a capability that is fundamental to more-complex time series reasoning tasks (Chow et al., 2024)—analogous to how addition is foundational for mathematical reasoning. Describing time series with language is also an important task on its own for both general-purpose (Trabelsi et al., 2025; Ito et al., 2025; Chow et al., 2024) and task-specific settings (Han et al., 2024; Martínez-Cruz et al., 2021; Serrano Chica, 2020; Martínez-Cruz et al., 2023). For example, Law et al. (2005) shows that neonatal ICU staff

---

[*]All of our code and data needed to reproduce our results will be made public.

made better treatment decisions when shown descriptions of physiological time series, which has recently been corroborated in finance (Yarushkina et al., 2025). However, we lack evaluations of whether state-of-the-art multi-modal time series models can describe time series.

To address these limitations, we introduce BEDTime, the first rigorous benchmark enabling direct, task-specific comparisons of language models (LLMs,VLMs, and TSLMs) for recognizing, distinguishing, and generating generic natural language descriptions of univariate time series. We propose that a successful multi-modal time series reasoning model should be able to perform at least three tasks: (1) *recognize* accurate descriptions when shown a corresponding time series, (2) *differentiate* correct descriptions from incorrect descriptions, and (3) *generate* language descriptions that capture key visual properties of an input time series. We create BEDTime by unifying four recent datasets, formatting them to support each of our proposed tasks and including evaluation metrics, resulting in 10,164 time series with corresponding descriptions.

Using BEDTime, we conduct extensive head-to-head comparisons of LLMs, VLMs, and recent pre-trained time series language models. On each task, we find that VLMs are surprisingly successful, despite being underutilized for multi-modal time series analysis. However, they still have significant room for improvement. We also find that prompting LLMs is largely unsuccessful on this foundational task, despite its popularity. These findings are also corroborated in a human evaluation. We also find that custom-trained TSLMs, frequently outperform LLMs of comparable size, indicating a clear avenue for future research. Finally, we observe that all models are susceptible to simple, realistic robustness tests, suggesting that surface-level accuracy may not truly reflect their ability to understand time series data.

In summary, our work contributes to the literature as follows:

1. We introduce BEDTime, a unified benchmark containing 10,164 unique time series with corresponding descriptions formatted for 3 tasks including image, numeric, and text representations.
2. Our evaluations using BEDTime suggest future works should prioritize visual representations of time series, recent methods are generally fragile to realistic perturbations, and there remains significant value in developing novel multi-modal time series—language architectures.
3. We introduce comprehensive evaluation methods, combining both human expert assessments and automated metrics, to rigorously measure model performance. Additionally, upon publication we publicly release all relevant code and data to facilitate transparent, reproducible, and further development of multi-modal time series reasoning models.

## 2 RELATED WORKS

**Benchmarks for Multi-Modal Time Series Reasoning.** Recent benchmarks for time series reasoning fall into three categories: synthetic datasets, empirical human-annotated datasets, and domain-specific Question-Answering datasets. Synthetic datasets such as TRUCE-Synthetic (Jhamtani & Berg-Kirkpatrick, 2021), TaxoSynth (Fons et al., 2024), and SUSHI (Kawaguchi et al., 2024) allow controlled evaluation over trends, periodicities and stochastic fluctuations but often lack the nuances of real world time series. TRUCE-Stock (Jhamtani & Berg-Kirkpatrick, 2021) is one of the few empirical datasets with real-world stock-data time series and crowd-authored descriptions. QA-style evaluations like TimeSeriesExam (Cai et al., 2024) assess reasoning via templates, while ECG-QA (Oh et al., 2023), DeepSQA (Xing et al., 2021), and PIXIU (Xie et al., 2023) target domain-specific question answering with time series as auxiliary input but often rely on machine-generated language that is not generalizable across domains or even across different datasets of the same domain. Moreover, existing benchmarks lack multimodal coverage, evaluation frameworks for open-ended description generation, and robustness testing under real-world perturbations; BEDTime includes all three, and releases all code and data upon acceptance to facilitate reproduction and evaluation of new datasets and models.

**Methods for Multi-Modal Time Series Reasoning.** Recent work on time series reasoning spans three paradigms: LLMs, VLMs, and recently TSLMs. (1) *LLMs:* LLMTIME (Gruver et al., 2023) treats forecasting as next-token prediction; PromptCast (Xue & Salim, 2023) frames it as a question-answering set-up. Several surveys (Zhang et al., 2024; Jiang et al., 2024) outline prompting and fine-

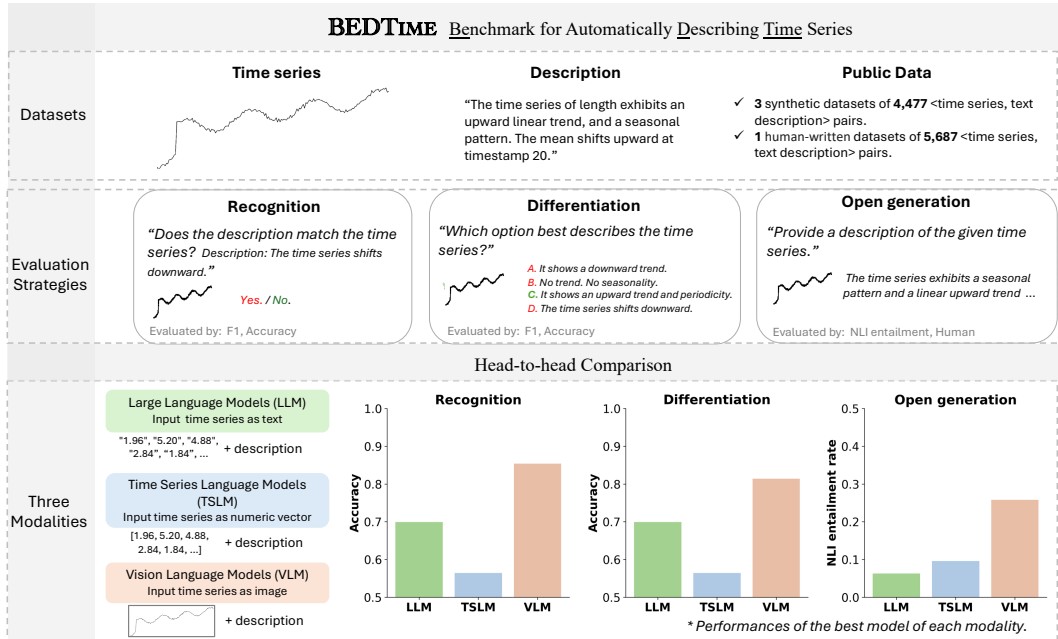

Figure 1: Overview of the benchmark for automatic time series description (BEDTime), featuring head-to-head comparisons across three modalities. The benchmark includes two parts: a diverse collection of public datasets containing time series paired with textual descriptions of their visual properties (first row); and evaluation strategies across three tasks: description recognition, description differentiation, and open-ended generation given a time series (second row). The benchmark is compared head-to-head across three modalities (third row).

tuning strategies, but highlight LLMs' limited numeric reasoning and cross-domain generalization. (2) *VLMs:* Models like CLIP (Radford et al., 2021), BLIP-2 (Li et al., 2023), and Time-VLM (Zhong et al., 2025) process plotted time series via vision-language alignment. These models capture trends visually but lose signal fidelity. (3) *TSLMs:* ChatTime (Wang et al., 2025) introduces a foundation model that discretizes raw time series and encodes them into instruction-style prompts, enabling LLMs to perform forecasting, classification, and question-answering with strong zero-shot generalization across modalities. ChatTS (Xie et al., 2025) leverages a synthetic data generator to align time series and text at scale, training a time-series-aware LLM that supports fine-grained reasoning and description generation over multivariate inputs; Trabelsi et al. (2025) uses cross-modal retrieval; CLaSP (Ito et al., 2025) learns joint embeddings; Chow et al. (2024) integrate CoT with lightweight encoders. These methods all target alignment, captioning, or reasoning, but lack fundamental comparative evaluation. BEDTime bridges these gaps with unified, multimodal evaluation of LLMs, VLMs, and TSLMs across description recognition, differentiation, and generation—revealing how models reason over text, images, and raw time series inputs respectively.

## 3 BEDTIME: A BENCHMARK FOR DESCRIBING TIME SERIES

As illustrated in Figure 1, we propose BEDTime, a benchmark to evaluate the degree to which models can recognize, differentiate, and generate descriptions of patterns in univariate time series data using language. BEDTime includes over 10,000 pairs of time series and descriptions, unified from four recent datasets and formatted for evaluation on three tasks that we propose. Each task is fundamental to more complex and desirable reasoning capabilities. By focusing on general, simple, and interpretable domain-agnostic language descriptions of time series' *visual* properties like trend, periodicity, and abrupt changes, BEDTime can be applied across models.

**Notation.** Consider a dataset $\mathcal{D} = \{(x_i, d_i)\}_{i=1}^N$ containing $N$ pairs of time series $x$ and corresponding descriptions $d$. Each time series $x_i \in \mathbb{R}^T$ is a real-valued sequence of $T$ values, which are described by the corresponding natural language description $d_i$. For example, Figure 1 shows a time series and its description. A multi-modal foundation model $f(\cdot)$ may take in a time series $x_i$ and a prompt $p_i$ and produce an output $f(x_i, p_i)$ corresponding to the task description in $p_i$. Section 3.1 details how $p_i$ and $f(x_i, p_i)$ vary based on the task being evaluated.

| Dataset | $N$ | Series Length | Description Word Length | Description Source | Time Series Source |
|---|---|---|---|---|---|
| TRUCE-Stock | 5687 | 12 | 2-9 | Human | Google Finance API |
| TRUCE-Synthetic | 1677 | 12 | 1-8 | Synthetic | Synthetic |
| TaxoSynth | 1400 | 24-150 | 5-40 | Synthetic | Synthetic |
| SUSHI | 1400 | 2048 | 5-60 | Synthetic | Synthetic |

Table 1: Overview of the four datasets in `BEDTime`, including the number of unique time series–text pairs, sequence lengths, and description characteristics.

## 3.1 PROPOSED TASKS

We propose three tasks that capable time series reasoning models should be able to perform. Tasks 1 and 2 are formulated as Question–Answering tasks, where models are given time series and asked to recognize or differentiate correct and incorrect descriptions. Task 3 is free-response, which is hard to evaluate, so we leverage an automated metric and conduct human evaluations.

**Description Recognition (Task 1).** Given a time series and an accompanying description, a model must determine whether the description is valid. We format this as a True/False task and design the prompt $p_i$ to encourage $f(x_i, d_i) \in \{\text{"True"}, \text{"False"}\}$. Because our datasets do not contain "incorrect" (False) time series and description pairs naturally, we construct them by sampling from within our data. We use four such methods to ensure the robustness of our benchmark to the choice of sampling method. One method assesses the similarity of captions, while the other three methods assess the similarity of time series. In all cases, we select the most dissimilar option as the incorrect description. Appendix B provides further details about our method for selecting incorrect options.

**Description Differentiation (Task 2).** Given a time series $x_i$, the model must select the correct description $d_i$ from a set of four options $\{d_i, d_j, d_k, d_l\}$. This is posed as a multiple-choice task, with prompt $p_i$ formatted to present the options and elicit a letter-valued prediction $f(x_i, p_i) \in \{A, B, C, D\}$, in which each letter corresponds to a description option. A prediction is correct if it corresponds to the index of the true description $d_i$. As in Task 1, incorrect options are generated by sampling the three most dissimilar descriptions from the dataset.

**Open Generation (Task 3).** Given a time series $x_i$, the model must generate a natural language description $d_i = f(x_i, p_i)$ based on a prompt $p_i$ (e.g., "describe the given time series"). We evaluate generation quality using both automatic and manual strategies. For automatic evaluation, we use a natural language inference (NLI) model to compute unidirectionally and bidirectionally entailment between generated and ground truth descriptions. The human evaluation criteria are adapted from prior literature on generating linguistic descriptions of time series (GLiDTS) work Martínez-Cruz et al. (2023). Full evaluation details are provided in Section 4.2.2.

## 3.2 DATASET DESCRIPTION AND PRE-PROCESSING

`BEDTime` composes four existing time series description datasets, each reformatted for the tasks above: TRUCE-Stock (short, noisy real-world stock data with crowd-sourced descriptions) and TRUCE-Synthetic (short simple synthetic data with both crowd-sourced and template-based descriptions) (Jhamtani & Berg-Kirkpatrick, 2021), TaxoSynth (variable-length taxonomic patterns based on critical aspects of frequently analyzed time series data with synthetic descriptions) (Fons et al., 2024), and SUSHI (long, complex signals with prominent trends, seasonality, and white noise and template-based descriptions) (Kawaguchi et al., 2024). These datasets are not originally designed for our specific task, so we perform several dataset-specific preprocessing steps to adapt them for use in our experiments. For TRUCE-Stock and TRUCE-Synthetic, when a time series had multiple valid annotations, we create a separate entry for each annotation–series pair. In the case of TaxoSynth, we begin with their ten datasets, each from a different taxonomy, and use the univariate time series. We use only the qualitative descriptions, which are concise, generalizable pattern summaries. To be consistent with other datasets and recent benchmarks (Cai et al., 2024; Gruver et al., 2023), we remove all timestamps, keeping only the value sequences. For both the SUSHI and TaxoSynth datasets, we retain class and subclass labels, which were used to select incorrect options to ensure that distractor descriptions for recognition and differentiation tasks always come from different classes and subclasses than the target. *Note: All code and data are included in the supplementary material and will be made publicly available upon acceptance.*

| Models | Datasets | | | | Rank |
|---|---|---|---|---|---|
| | TRUCE-Stock | TRUCE-Synth | SUSHI | TaxoSynth | |
| **LLMs** | | | | | |
| GPT-4o | .64 | .80 | .74 | .73 | 2.25 |
| GPT-4o+LT | **.66** | **.81** | **.84** | **.79** | **1.00** |
| Gemini-2.0 | .60 | .72 | .72 | .69 | 3.62 |
| Llama-3.1-8B | .50 | .54 | .49 | .51 | 6.75 |
| Qwen2.5-14B | .61 | .72 | .51 | .59 | 4.25 |
| Qwen2.5-7B | .65 | .63 | .51 | .57 | 4.50 |
| Phi-3.5 | .59 | .52 | .54 | .57 | 5.62 |
| **VLMs** | | | | | |
| GPT-4o | **.75** | **.83** | **.96** | **.88** | **1.00** |
| Gemini-2.0 | .68 | .72 | .86 | .78 | 2.38 |
| Qwen2.5-VL-7B | .61 | .72 | .61 | .73 | 3.38 |
| Phi-3.5-vision | .69 | .66 | .79 | .68 | 3.25 |
| **TSLMs** | | | | | |
| ChatTime-7B | .10 | .13 | .28 | .25 | 2.00 |
| ChatTS-14B | **.42** | **.60** | **.78** | **.77** | **1.00** |

| Models | Datasets | | | | Rank |
|---|---|---|---|---|---|
| | TRUCE-Stock | TRUCE-Synth | SUSHI | TaxoSynth | |
| **LLMs** | | | | | |
| GPT-4o | .52 | .73 | .65 | .71 | 2.50 |
| GPT-4o+LT | **.56** | **.79** | **.67** | **.75** | **1.00** |
| Gemini-2.0 | .53 | .63 | .63 | .61 | 3.12 |
| Llama-3.1-8B | .26 | .28 | .23 | .26 | 7.00 |
| Qwen2.5-14B | .53 | .68 | .45 | .53 | 3.38 |
| Qwen2.5-7B | .43 | .47 | .24 | .35 | 5.50 |
| Phi-3.5 | .32 | .37 | .44 | .50 | 5.50 |
| **VLMs** | | | | | |
| GPT-4o | **.67** | **.81** | **.97** | **.81** | **1.00** |
| Gemini-2.0 | .65 | .75 | .90 | .71 | 2.25 |
| Qwen2.5-VL-7B | .62 | .73 | .87 | .76 | 2.75 |
| Phi-3.5-vision | .61 | .71 | .76 | .70 | 4.00 |
| **TSLMs** | | | | | |
| ChatTime-7B | .07 | .10 | .24 | .15 | 2.00 |
| ChatTS-14B | **.41** | **.59** | **.60** | **.65** | **1.00** |

(a) Recognition          (b) Differentiation

Table 2: Accuracy of LLMs, VLMs, and TSLMs on recognition and differentiation tasks. LT denotes the LLMTime prompting strategy Gruver et al. (2023). DTW distance is used to pick dissimilar descriptions. Reported ranks are intra-modal. For additional results, refer to Appendix D.

## 4 EXPERIMENTS

### 4.1 EXPERIMENTAL SETUP

**Models.** We evaluate six LLMs, four VLMs, and two TSLMs on BEDTime. This includes proprietary as well as public models that range from 4.2B to 14B parameters.

**LLMs.** We evaluate GPT-4o (OpenAI et al., 2024), Gemini-2.0-Flash (Google, 2025), LLaMA-3.1-8B-Instruct (Grattafiori et al., 2024), Phi-3.5-Mini-Instruct (Abdin et al., 2024), Qwen2.5-14B-Instruct-1M, and Qwen2.5-14B-Instruct-1M (Yang et al., 2025). We convert time series to strings of comma-separated values, following prior works (Gruver et al., 2023; Cai et al., 2024). For GPT-4o, we also compare with the LLMTime (Gruver et al., 2023) prompting strategy, which we denote as LT.

**VLMs.** We evaluate GPT-4o-Vision (OpenAI et al., 2024), Gemini-2.0-Flash-Vision (Google, 2025), Qwen2.5-VL-7B-Instruct (Qwen et al., 2025), and Phi-3.5-Vision-Instruct (Abdin et al., 2024). Here, we represent the time series as matplotlib-rendered plots.

**TSLMs.** We evaluate ChatTS (Xie et al., 2025) and ChatTime (Wang et al., 2025), which operate on time series inputted directly as numerical vectors.

**Metrics.** We use three metrics and human evaluations. Recognition and Differentiation are classification tasks with balanced datasets, so we measure accuracy. Open Generation is harder to evaluate because the models generate freeform text descriptions. Therefore, we take two approaches: First, we use automated metrics to compute similarity between generated responses. Second, we conduct a human evaluation with three annotators for six criteria adapted from prior literature (Martínez-Cruz et al., 2023). Further details on open generation evaluations are available in Section 4.2.2.

### 4.2 RESULTS

#### 4.2.1 CAN FOUNDATION MODELS RECOGNIZE AND DIFFERENTIATE TIME SERIES DESCRIPTIONS?

We first compare all models on Recognition and Differentiation because they are both classification tasks with balanced datasets evaluated using accuracy. For both tasks, to capture general trends, we also report each model's average rank compared to all models of the same modality, on all datasets. Our main results for both tasks are found in Table 2, which is broken down into Recognition (Table 2a) and Differentiation (Table 2b).

Overall, we observe that VLMs are consistently the best-performing models across each dataset and task, and GPT-4o is the best VLM. This is expected, as descriptions largely depend on visual properties, necessitating vision encoders. However, on these datasets, even the best VLM is surprisingly inaccurate, at these extremely simple tasks. We posit that a successful time series reasoning model should near 100% accuracy for both description recognition and differentiation, so there is clear room for further method development on these task. LLMs are second best, with some of the best models rivaling VLMs in some cases (e.g., GPT-4o on TRUCE-Synth). GPT-4o also benefits from LLMTime prompting strategy, though only slightly. Notably, Qwen2.5-14B is the best public model, suggesting a possible starting place for further development of public models. TSLMs appear to be the worst of the three options so far, though ChatTS outperforms LLMs of comparable parameter size on the

| Model | Natural language inference (NLI) entailment | | | Human evaluation on 6 designed criteria | | | | | |
|---|---|---|---|---|---|---|---|---|---|
| | Bi-directional | Gen → GT | GT → Gen | Valid | Patterns | Position | Noise | Step-by-Step | Concise |
| **GPT-4o Vision** | **14.41** | **47.65** | **15.29** | 1.00 | 0.891 | 0.751 | 0.815 | 0.853 | 0.851 |
| **GPT-4o Text** | 2.94 | 9.71 | 6.18 | 1.00 | 0.361 | 0.234 | 0.624 | 0.681 | 0.715 |
| **ChatTS** | 2.65 | 23.53 | 2.65 | 1.00 | 0.738 | 0.540 | 0.657 | 0.613 | 0.620 |

Table 3: Evaluation results for time series descriptions from three models, showing entailment percentages (left) and averaged human evaluation scores (right). We assess the rate at which the LM description entails the ground truth (Gen → GT), the ground truth entails the LM description (GT → Gen), and when both entail each other (bi-directional entailment). For additional analysis and metrics, see Appendix G.

TRUCE-Synth, SUSHI, and TaxoSynth datasets. Interestingly, ChatTime severely underperforms in each case, primarily due to refusals to respond rather than incorrect selections (see Appendix F for further analysis).These trends hold for both Recognition and Differentiation task, as shown in Table 2a and Table 2b respectively, where the models are ranked similarly for both tasks.

Although VLMs dramatically outperform LLMs, their architectures still incorporate their base LLM components. To better isolate the contribution of their visual encoders, we compare each VLM directly with its LLM counterpart. Across datasets and tasks, VLMs of all sizes and model families—both proprietary and public—consistently surpass their LLM baselines, irrespective of prompting strategies, underscoring the importance of visual representations for description recognition and differentiation, as shown in Figure 4. In comparison, increasing model scale—e.g., doubling LLM parameters from 7B to 14B—offers only marginal improvements, and switching from strong open-source LLMs to proprietary ones results in similarly modest gains. Notably, simple number formatting via LLMTIME can yield larger benefits than either scaling or switching model families, underscoring the importance of input representation over raw model size. While general-purpose LLMs lag behind on long or structured sequences, well-trained TSLMs can match—or even surpass—text-only models twice their size, particularly on datasets like SUSHI and TaxoSynth that reward temporal inductive biases. Nevertheless, VLMs still outperform TSLMs in the vast majority of settings, confirming that visual context remains a key strength across time series data input modalities.

Some models fail to produce outputs in the necessary which we as errors, leading accuracy to appear below random, as discussed further in Appendix F. We count such cases as errors, which can lead accuracy to appear below random. Full per-dataset results for the four annotation distractor selection schemes are presented in Appendix D, demonstrating that performance trends remain consistent across strategies—reinforcing the robustness of our findings.

### 4.2.2 CAN FOUNDATION MODELS GENERATE TIME SERIES DESCRIPTIONS?

We next evaluate the open-generation capabilities of the top-performing models from each modality on Tasks 1 and 2: GPT-4o (for LLMs), GPT-4o-Vision (for VLMs), and ChatTS-14B (for TSLMs). We use the SUSHI dataset because it has long and richly-annotated descriptions, complex temporal structure (including trend, seasonality, and noise), and long sequence length (2048 steps).

Given a time series, each model is prompted to generate a natural language description capped at 150 tokens. Example prompts are included in Appendix C. We evaluate outputs using both automated and human-centered metrics. Results are summarized in Table 3 and discussed below.

**Automatic Evaluation.** We use a natural language inference (NLI) model (`deberta-base-long-nli`) (Sileo, 2024) to compute entailment between generated and ground truth descriptions. Entailment is assessed bidirectionally (Machine Generated → Ground Truth and Ground Truth → Machine Generated), and we also report the proportion of strict mutual entailment. As shown in Table 3 (left), GPT-4o-Vision achieves the highest entailment rates, followed by GPT-4o-Text and ChatTS. Notably, entailment from generation to ground truth is consistently higher than the reverse, suggesting that models tend to produce more specific or detailed descriptions than those in the reference set.

**Human Evaluation.** We also conduct a human evaluation to assess whether generated descriptions are generic, accurate, interpretable, and fluent. We selected 340 model-generated descriptions using stratified random sampling over the entire SUSHI dataset, ensuring that time series from all classes and subclasses were proportionally represented. Three annotators then scored these model-generated description (40 overlapping for agreement, 100 unique per annotator), across three modalities, using **six** binary criteria: (1) *Coherence*: Descriptions must contain logically connected and meaningful statements. (2) *Pattern Identification*: Relevant features (e.g., trends, spikes, periodicity) are correctly

recognized. (3) *Temporal Localization*: The positions of these features within the sequence are correctly specified. (4) *Noise Filtering*: Descriptions omit insignificant variations or noise. (5) *Abstraction*: Descriptions are high-level, avoiding overly detailed step-by-step repetition of the input sequence. (6) *Linguistic Quality*: Generated langyuage is concise, fluent, and semantically generic. All models successfully produced relevant and coherent outputs, but GPT-4o-Vision received the highest scores for identifying and localizing temporal patterns. GPT-4o-Text produced fluent language but often misstated underlying trends, while ChatTS generated structurally accurate but sometimes overly literal or low-abstraction descriptions. Inter-annotator agreement was high, with Cohen's Kappa and Krippendorff's Alpha exceeding 0.8 for all criteria.

### 4.2.3 How robust are foundation models against realistic perturbations?

We next evaluate how robust model performance on `BEDTime` is to realistic perturbations—an essential consideration for any system claiming to "describe or "understand" time series. In real-world deployments, sequences may vary in length, or exhibit missing values, that challenge descriptive precision. To this end, we conduct **five** robustness tests designed to probe a models' generalization capabilities beyond the clean, fully observed inputs used in our main benchmark.

First, we assess robustness to increased sequence length by linearly interpolating time series—preserving core properties while increasing resolution. This allows us to evaluate whether model descriptions remain consistent as inputs become longer. Second, we introduce structured missingness via uniform random masking, simulating data sparsity that is common in practical settings. We additionally evaluate the effects of additive Gaussian noise by applying nine levels of perturbation (measured by standard deviation) and amplitude scaling on model performance at five levels, where each level corresponds to a scalar multiplier applied to every element of the time series. Lastly we also test the effect of chain of thought prompting strategy (Wei et al., 2023) on LLM performance.

**Effect of Varying the Length of the Time Series.** We observe that recognition and differentiation accuracy for text-only LLMs declines as sequence length increases (see Figure 2A and 2B) —a trend consistent with known limitations of Transformer architectures. This degradation stems from three key factors: (1) the $\mathcal{O}(L^2)$ cost of self-attention with respect to input length $L$, which limits usable context via truncation or approximation (Vaswani et al., 2023); (2) attention diffusion, wherein softmax weights flatten over longer inputs, reducing focus on salient time steps; and (3) poor generalization of positional encodings beyond pretraining lengths, which distorts temporal localization (Gruver et al., 2023; Fons et al., 2024; Merrill et al., 2024). These limitations are most pronounced in smaller open-source models like LLaMA, which exhibit sharp drops in performance with input length, whereas larger models such as GPT-4o and Gemini decline more gradually. Prior work further shows that such degradation is amplified by position bias—especially when target values appear late in the sequence—and by the inability of standard tokenizations to preserve long-range temporal structure. In contrast, both `ChatTS` and `ChatTime` improve with increasing sequence length, suggesting that TSLMs benefit from interpolation and are more robust to time series longer inputs.

Interpolating time series increases floating-point precision, which drastically inflates token counts for LLMs. This raises a natural question: is the observed performance drop primarily due to tokenization overhead rather than sequence length itself? To test this, we scaled interpolated values by 100 to convert floats into integers and evaluated whether this improved model accuracy. From Figure 5 we see that scaling does aid model performance across all models except Qwen2.5-14B, whose performance declines. We attribute this to Qwen2.5-14B's attention diffusion and weaker reliance on discrete token structure: its larger key-value head count and higher parameter capacity encourage more distributed representations, though the model shows limited gains on number-sensitive tasks (Yang et al., 2025).

**Effect of Missing Data.** We next measure the effect of missing values on model performance. To simulate real-world data sparsity, we introduce missingness by uniformly at random masking out a fraction of each time series—replacing those entries with NaN—at four levels (5 %, 25 %, 50 %, and 75 %). We then evaluate our same suite of language-only models—including proprietary and open-source LLMs—on both Recognition and Differentiation tasks under each missingness condition. The results are shown in Figure 2C. Missing values degrade performance across all models and datasets. Accuracy remains relatively stable up to 25% data missingness, particularly for LLMs, but

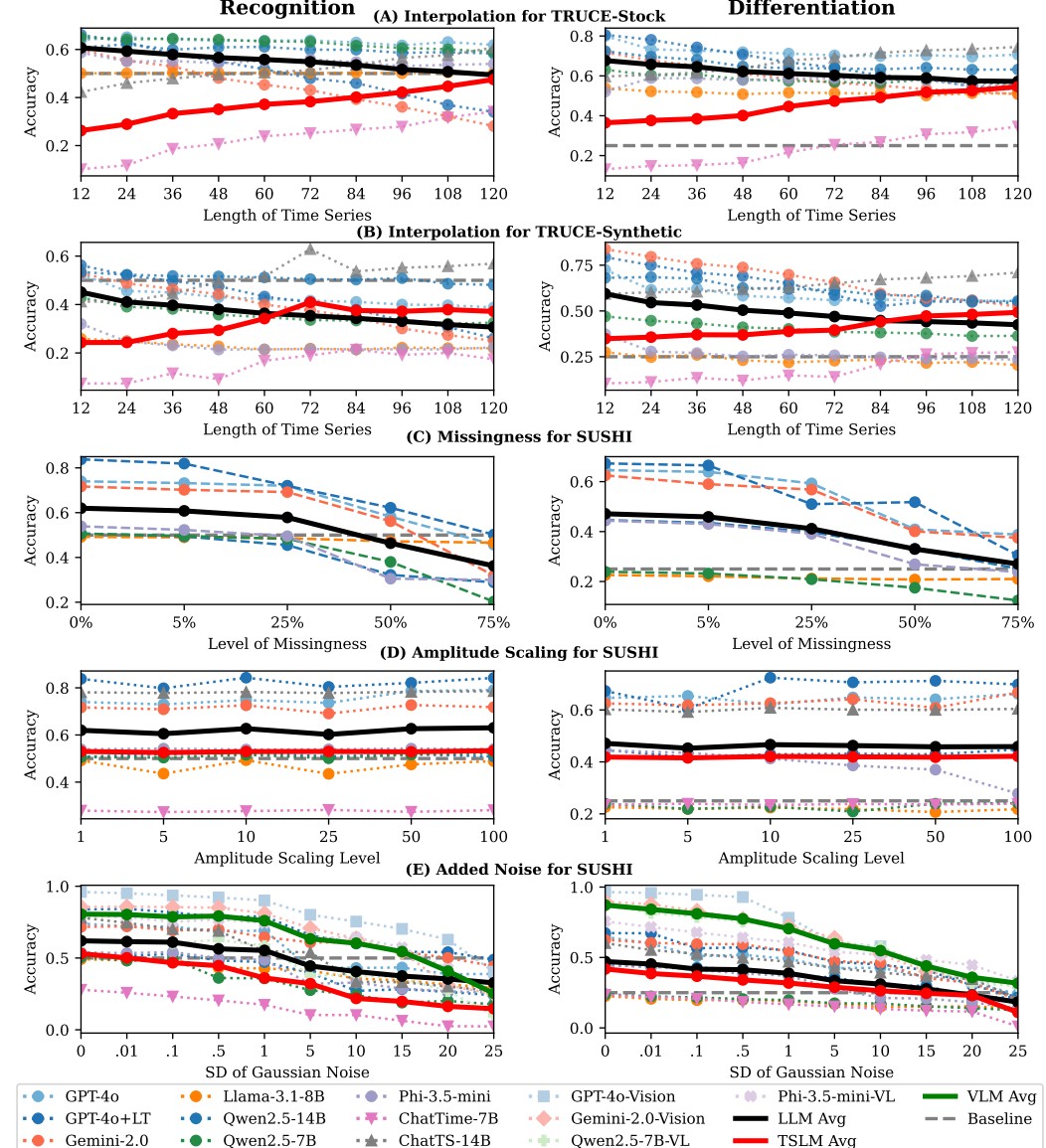

Figure 2: Robustness of LLMs and TSLMs to sequence length, missingness, added gaussian noise and amplitude scaling. Recognition and differentiation accuracy decline for LLMs as time series grow longer or contain more missing values—particularly beyond 50. Also as the signals get noisier performance decline however models are relatively robust to amplitude scaling. DTW distance is used to pick dissimilar descriptions.

drops sharply once 50% or more of the time series is masked. This trend holds for both recognition and differentiation tasks.

**Effect of Amplitude Scaling.** We evaluate the effect of amplitude scaling on model performance by multiplying each time series by a scalar at five levels (5, 10, 25, 50, 100). We then evaluate our suite of models on both Recognition and Differentiation tasks under each scaling condition. The results are shown in Figure 2D. Both LLMs and TSLMs are broadly robust to amplitude scaling, with only minor changes in accuracy across levels. Notably, LLMs show slight improvements at the highest scaling factor, likely due to enhanced visibility of features in scaled signals. This trend holds across both recognition and differentiation tasks.

**Effect of Gaussian Noise.** We measure the effect of additive Gaussian noise on model performance by perturbing each time series with noise at nine levels (0, 0.01, 0.1, 0.5, 1, 5, 10, 15, 20, 25), defined by the standard deviation of the noise distribution. We then evaluate our suite of models—including LLMs, VLMs, and TSLMs—on both Recognition and Differentiation tasks under each noise condition. The results are shown in Figure 2E. Performance consistently degrades as

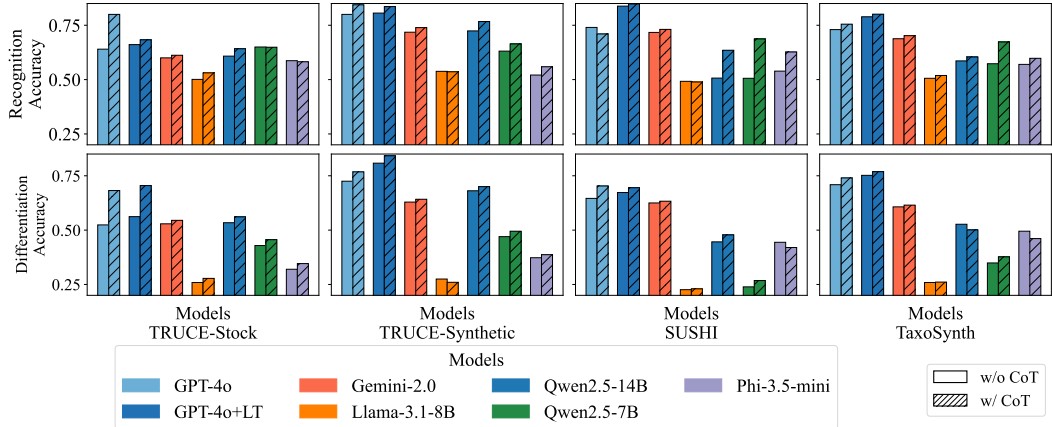

Figure 3: Impact of Chain-of-Thought (CoT) prompting on language-only models' accuracy across four datasets and two tasks. CoT consistently improves both recognition and differentiation performance, with the largest gains observed on the differentiation task—especially for proprietary models. DTW distance is used to pick dissimilar descriptions.

noise increases. Text-only models' performance rapidly falls below task-specific random baselines, whereas multimodal models (VLMs and TSLMs) show greater robustness, with slower degradation under higher levels of additive Gaussian noise.

**Effect of Chain-of-Thought.** We evaluate the impact of Chain-of-Thought (CoT) prompting on model performance by testing all text only models from our model suite on both Recognition and Differentiation tasks, with and without CoT prompting across all four datasets. The results are shown in Figure 3. Across every model and dataset, Chain-of-Thought prompting yields a clear and consistent uplift. Text-only language models benefit most markedly, especially on the harder Differentiation task, where CoT closes much of the gap to vision-language models. Proprietary systems show larger relative gains than their open-source counterparts. These findings demonstrate that CoT is a simple yet effective strategy for improving language-only models in time series understanding.

## 5 CONCLUSIONS

We introduce `BEDTime`, a unified benchmark for evaluating language models on the task of time series description—a fundamental capability for temporal reasoning. `BEDTime` defines three new tasks and adapts four recent datasets to evaluate language, vision–language, and time series–language models. Our experiments show that models across modalities, families, and sizes still have clear room for improvement on every task, especially under real-world perturbations. At the same time, VLMs demonstrate notable successes, underscoring the value of visual representations for time series understanding. Overall, `BEDTime` provides a unified and extensible framework for evaluating time series description across three modalities; upon acceptance, we will release its code and data to ensure reproducibility and support seamless evaluation of future models and emerging time series–text datasets.

## 6 LIMITATIONS

`BEDTime` has several limitations that open promising directions for future work. It currently supports only univariate time series, leaving multivariate extensions unexplored. Much of the data is synthetic, with only TRUCE-Stock reflecting real-world signals and only TRUCE-Stock and TRUCE-Synthetic providing human-annotated descriptions—both limited in length and richness, motivating broader curation of diverse and complex datasets. Our evaluations also depend on general-purpose NLI models which introduce domain mismatch and numerical sensitivity, highlighting the need for time series–specific NLI metrics or standardized automated evaluation protocols. Moreover, reliance on instruction-tuned models risks conflating instruction-following with reasoning, underscoring the value of instruction-robust task designs. Addressing these limitations offers clear pathways for extending `BEDTime` into a richer and more versatile framework for future research.

ETHICS STATEMENT

This work introduces `BEDTime`, a benchmark for evaluating models on the foundational task of describing univariate time series. Our study does not involve human subjects, personal data, or sensitive information. Among the four datasets in `BEDTime`, TRUCE-Synthetic contains synthetic time series paired with a mix of non-identifiable, crowd-authored and template-based descriptions, while TaxoSynth and SUSHI are fully synthetic in both their signals and captions. In contrast, TRUCE-Stock is composed of real-world financial time series with non-identifiable, crowd-authored descriptions. Notably, all of our data is drawn from previously released research datasets that are already publicly available and open-source. Human evaluations were conducted by members of our research group to assess the quality of model-generated descriptions. These annotators were not study participants but collaborators, and no identifiable or sensitive data was involved, so IRB approval was not required. The benchmark is intended solely for research purposes, and we emphasize that potential downstream use in sensitive domains should be carefully evaluated. Importantly, while our experiments benchmark large language, vision–language models and time series and language models, no AI or LLM systems were used in writing this manuscript or in developing the accompanying code.

REPRODUCIBILITY STATEMENT

We have made extensive efforts to ensure reproducibility. All experiments are described in detail in the main manuscript and Appendix, including dataset pre-processing steps, task formulations, evaluation metrics, model configurations and implementation details. For submission, we provide a zip file of our code and processed datasets as supplementary material. Upon acceptance, we will release the full codebase and preprocessed `BEDTime` datasets publicly, accompanied by detailed user documentation to facilitate seamless use and extension.

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

# APPENDIX

## A  DATASET DESCRIPTION

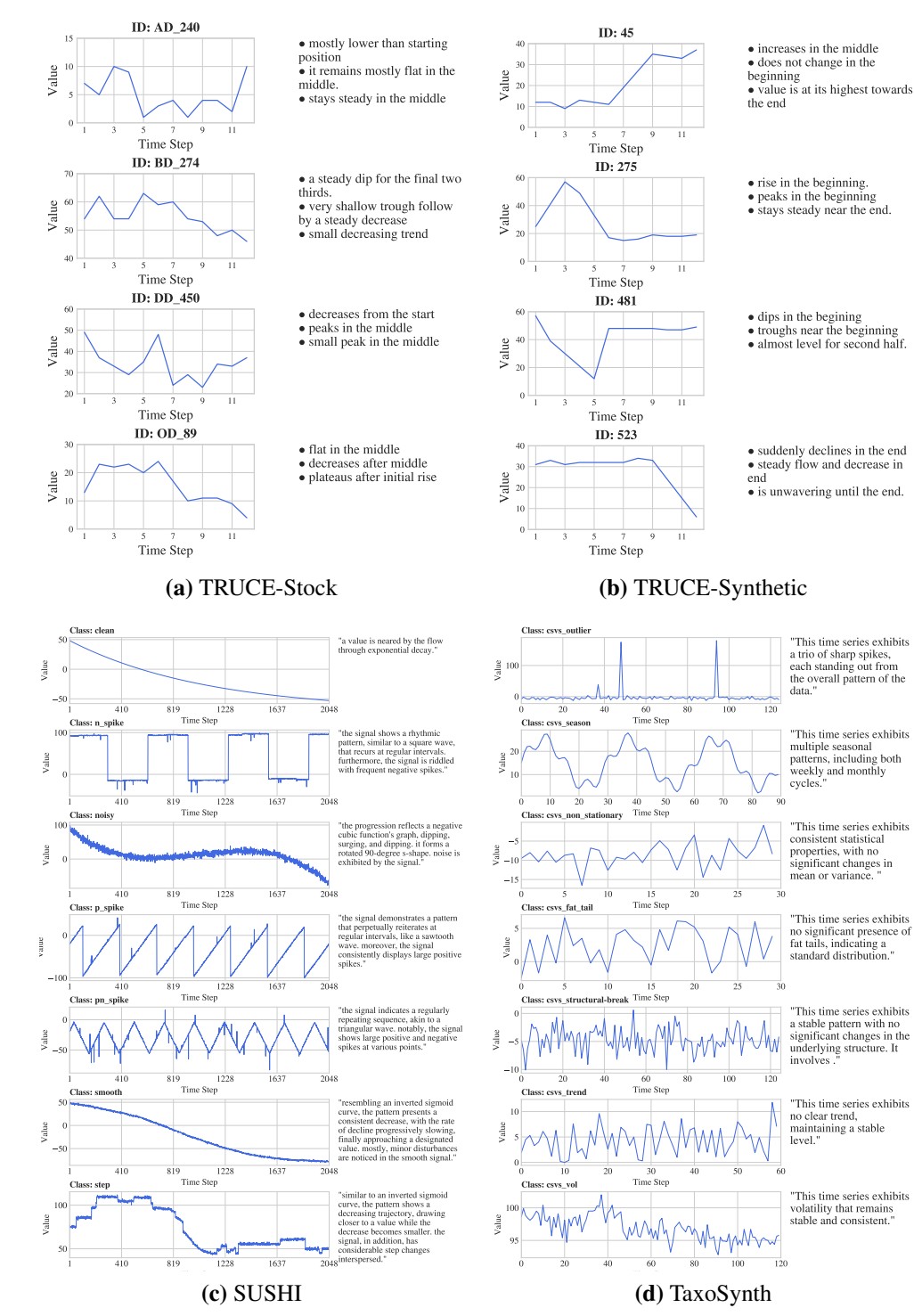

**(a)** TRUCE-Stock

**(b)** TRUCE-Synthetic

**(c)** SUSHI

**(d)** TaxoSynth

Table 4: Representative time series and descriptions from each dataset used in `BEDTime`. These examples illustrate variations in sequence length, description style, and data generation method.

To support our benchmark tasks, we unify and reformat four recent datasets containing paired time series and natural language descriptions. These datasets vary in realism, sequence length, and linguistic complexity, and are summarized in Table 1. These datasets described in Section 3.2 support the recognition, differentiation, and generation tasks, though each dataset is used selectively depending on task requirements and annotation structures.

Table 4 illustrates representative samples from each dataset, showcasing the diversity in time series patterns and caption styles—from concise, crowd-sourced descriptions of financial data to longer, templated descriptions of synthetically generated sequences. Together, the table and examples highlight the breadth of data modalities and linguistic forms evaluated by `BEDTime`.

## B    SAMPLING INCORRECT OPTIONS

To support both True/False and Multiple Choice formats in the Recognition and Differentiation tasks, we construct contrastive examples by selecting negative descriptions using four distinct strategies:

- **Caption-based similarity (Sentence-BERT):** We compute cosine similarity over Sentence-BERT embeddings and select descriptions that are semantically dissimilar to the reference.
- **Dynamic Time Warping (DTW):** We measure alignment costs between time series and choose those with the highest DTW distance from the input.
- **Euclidean Distance:** We identify point-wise dissimilar series based on maximum L2 distance.
- **Longest Common Subsequence (LCSS):** We retrieve sequences with minimal overlapping subsequences, prioritizing structural dissimilarity.

Only the Sentence-BERT strategy operates over natural language annotations; the other three assess distance directly in time series space. When multiple annotations exist for a given time series, we randomly sample one for evaluation. Negative samples are selected to be maximally dissimilar, simplifying the contrastive setup and providing an upper-bound estimate of model performance. This design ensures that the benchmark evaluates models' ability to reject clearly incorrect options before advancing to more fine-grained reasoning.

## C    PROMPTS

We include the exact prompts used for each `BEDTime` task below. These prompts were used across models for consistent evaluation and follow the formats outlined in Section 3.1.

**Task 1: Recognition.**

> You are tasked with verifying if the provided annotation accurately describes the given time series.
> Please follow these instructions carefully:
> 1. Review the annotation: {description}.
> 2. Analyze the time series: {series}.
> 3. Determine if the annotation precisely matches the pattern depicted in the time series.
> Respond with **True** if the annotation accurately describes the time series.
> Respond with **False** if it does not. Avoid providing any additional comments or explanations.

**Task 2: Differentiation.**

> Carefully analyze the given time series and choose the single best option that most accurately describes its pattern.
> Follow these rules strictly:
> 1. Read all options before deciding.
> 2. Only output the chosen option, highlighted as A, B, C, or D.

3. Avoid adding extra text or explanations.
**Time series:** `{series}`
**Options:**

- A: `{option_1}`
- B: `{option_2}`
- C: `{option_3}`
- D: `{option_4}`

**Task 3: Open Generation.**

You are tasked with generating a textual description of the visual properties of the provided time series.
Please follow these instructions carefully:
1. Analyze the given time series data: `{series}`.
2. Identify and describe the most prominent visual features or patterns observed in the time series.
Consider characteristics such as trends, seasonality, anomalies, or significant changes.
Your response should be a concise textual description of the most pronounced visual properties of the time series.
Avoid including unnecessary details or unrelated commentary.

**Note:** The format of the time series input varies based on model modality. LLMs receive a comma-separated string of numeric values. VLMs receive a `matplotlib`-rendered image of the time series or a base64-encoded image string of the same. TSLMs are provided with the raw sequence as a `NumPy` array or Python list of floats.

# D    RECOGNITION AND DIFFERENTIATION: FULL RESULTS ACROSS ALL SAMPLING STRATEGIES

To support the findings presented in Sections 4.1 and 4.2, we report full accuracy and F1 score (F1 weighted in the case of differentiation) across all datasets and all four negative sampling strategies (Sentence-BERT, DTW, Euclidean, and LCSS) for the *Recognition* (True/False) and *Differentiation* (Multiple Choice) tasks. Tables 5, 6, 7, 8, 9, 10, 11, and 12 provide per-dataset, per-sampling performance scores.

| Model | S-Bert Embeddings with Cosine Similarity | | DTW Distance | | Euclidean Distance | | Longest Common Subsequence | |
|---|---|---|---|---|---|---|---|---|
| | Accuracy | F1 Score | Accuracy | F1 Score | Accuracy | F1 Score | Accuracy | F1 Score |
| **Language Models** | | | | | | | | |
| GPT-4o | 0.610 | 0.581 | 0.640 | 0.556 | 0.592 | 0.678 | 0.592 | 0.527 |
| GPT-4o (LLMTime) | **0.703** | **0.700** | **0.661** | **0.661** | **0.692** | **0.679** | **0.616** | **0.614** |
| Gemini-2.0-Flash | 0.607 | 0.546 | 0.600 | 0.528 | 0.585 | 0.557 | 0.534 | 0.472 |
| Llama-3.1-8B-Instruct | 0.501 | 0.334 | 0.501 | 0.334 | 0.501 | 0.334 | 0.501 | 0.334 |
| Qwen2.5-14B-Instruct-1M | 0.642 | 0.615 | 0.608 | 0.587 | 0.606 | 0.585 | 0.522 | 0.512 |
| Qwen2.5-7B-Instruct-1M | 0.674 | 0.656 | 0.650 | 0.635 | 0.643 | 0.630 | 0.519 | 0.516 |
| Phi-3.5-mini-instruct | 0.526 | 0.389 | 0.587 | 0.519 | 0.520 | 0.386 | 0.519 | 0.385 |
| **Vision-Language Models** | | | | | | | | |
| GPT-4o (with image inputs) | **0.842** | **0.840** | **0.750** | **0.719** | **0.760** | **0.727** | **0.840** | **0.820** |
| Gemini-2.0-Flash (with image inputs) | 0.793 | 0.786 | 0.682 | 0.648 | 0.683 | 0.648 | 0.675 | 0.648 |
| Qwen2.5-VL-7B-Instruct | 0.713 | 0.695 | 0.609 | 0.539 | 0.609 | 0.539 | 0.573 | 0.512 |
| Phi-3.5-vision-instruct | 0.824 | 0.822 | 0.686 | 0.685 | 0.679 | 0.679 | 0.612 | 0.611 |
| **Language Models for Time Series** | | | | | | | | |
| ChatTime-7B-Chat | 0.050 | 0.049 | 0.102 | 0.099 | 0.099 | 0.094 | 0.093 | 0.091 |
| ChatTS-14B | **0.414** | **0.393** | **0.423** | **0.404** | **0.405** | **0.398** | **0.453** | **0.438** |

Table 5: Performance of Various Language and Vision-Language Models on TRUCE-Stock Dataset in the Recognition Setting using 4 Different Negative Sampling Techniques

Table 5 reports model performance on the TRUCE-Stock dataset in the Recognition setting, using all four contrastive sampling strategies. As the only real-world dataset in the benchmark, TRUCE-Stock presents the greatest challenge, with lower overall performance across models. GPT-4o with

LLMTime performs best among language-only models, and GPT-4o with image input leads across all modalities, though with a smaller margin than in synthetic datasets. Most open-source LLMs underperform across distractor types, with particularly poor results from smaller models such as Phi and LLaMA. Time-series-specific models show limited effectiveness here as well, with ChatTS-14B underperforming relative to its base LLM counterpart, suggesting that its strengths may not translate as well to short, real-world financial sequences.

| Model | S-Bert Embeddings with Cosine Similarity | | DTW Distance | | Euclidean Distance | | Longest Common Subsequence | |
|---|---|---|---|---|---|---|---|---|
| | Accuracy | F1 Score | Accuracy | F1 Score | Accuracy | F1 Score | Accuracy | F1 Score |
| **Language Models** | | | | | | | | |
| GPT-4o | 0.790 | 0.760 | 0.800 | 0.792 | 0.691 | 0.654 | 0.780 | 0.776 |
| GPT-4o (LLMTime) | **0.806** | **0.804** | **0.806** | **0.802** | **0.696** | **0.695** | **0.802** | **0.802** |
| Gemini-2.0-Flash | 0.724 | 0.699 | 0.718 | 0.711 | 0.675 | 0.669 | 0.707 | 0.703 |
| Llama-3.1-8B-Instruct | 0.535 | 0.409 | 0.538 | 0.412 | 0.537 | 0.411 | 0.522 | 0.400 |
| Qwen2.5-14B-Instruct-1M | 0.819 | 0.817 | 0.724 | 0.724 | 0.710 | 0.710 | 0.715 | 0.715 |
| Qwen2.5-7B-Instruct-1M | 0.758 | 0.756 | 0.631 | 0.631 | 0.623 | 0.622 | 0.659 | 0.659 |
| Phi-3.5-mini-instruct | 0.588 | 0.519 | 0.521 | 0.386 | 0.581 | 0.514 | 0.538 | 0.482 |
| **Vision-Language Models** | | | | | | | | |
| GPT-4o (with image inputs) | **0.893** | **0.890** | **0.830** | **0.838** | **0.880** | **0.800** | **0.920** | **0.868** |
| Gemini-2.0-Flash (with image inputs) | 0.741 | 0.735 | 0.724 | 0.716 | 0.752 | 0.751 | 0.723 | 0.712 |
| Qwen2.5-VL-7B-Instruct | 0.607 | 0.537 | 0.716 | 0.699 | 0.717 | 0.699 | 0.660 | 0.648 |
| Phi-3.5-vision-instruct | 0.722 | 0.720 | 0.658 | 0.634 | 0.660 | 0.637 | 0.673 | 0.653 |
| **Language Models for Time Series** | | | | | | | | |
| ChatTime-7B-Chat | 0.174 | 0.172 | 0.132 | 0.130 | 0.132 | 0.130 | 0.162 | 0.160 |
| ChatTS-14B | **0.509** | **0.491** | **0.597** | **0.591** | **0.518** | **0.512** | **0.460** | **0.457** |

Table 6: Performance of Various Language and Vision-Language Models on TRUCE-Synthetic Dataset in the Recognition Setting using 4 Different Negative Sampling Techniques

Table 6 presents Recognition performance on the TRUCE-Synthetic dataset. As a structured synthetic benchmark with short sequences and well-defined up/down patterns, this dataset yields higher overall model performance than TRUCE-Stock. GPT-4o with LLMTime again leads among LLMs, while GPT-4o with image input achieves the highest scores across all distractor types, including near-perfect F1. ChatTS-14B performs notably better here than on TRUCE-Stock, highlighting its alignment with synthetic pattern recognition.

| Model | S-Bert Embeddings with Cosine Similarity | | DTW Distance | | Euclidean Distance | | Longest Common Subsequence | |
|---|---|---|---|---|---|---|---|---|
| | Accuracy | F1 Score | Accuracy | F1 Score | Accuracy | F1 Score | Accuracy | F1 Score |
| **Language Models** | | | | | | | | |
| GPT-4o | 0.874 | 0.838 | 0.740 | 0.700 | 0.764 | 0.701 | 0.862 | 0.860 |
| GPT-4o (LLMTime) | **0.894** | **0.876** | **0.838** | **0.773** | **0.813** | **0.722** | **0.885** | **0.882** |
| Gemini-2.0-Flash | 0.775 | 0.765 | 0.717 | 0.647 | 0.694 | 0.615 | 0.809 | 0.779 |
| Llama-3.1-8B-Instruct | 0.509 | 0.362 | 0.492 | 0.354 | 0.493 | 0.354 | 0.504 | 0.356 |
| Qwen2.5-14B-Instruct-1M | 0.510 | 0.357 | 0.507 | 0.357 | 0.506 | 0.354 | 0.510 | 0.358 |
| Qwen2.5-7B-Instruct-1M | 0.507 | 0.352 | 0.506 | 0.351 | 0.508 | 0.354 | 0.506 | 0.352 |
| Phi-3.5-mini-instruct | 0.556 | 0.484 | 0.539 | 0.472 | 0.567 | 0.492 | 0.582 | 0.503 |
| **Vision-Language Models** | | | | | | | | |
| GPT-4o (with image inputs) | **0.980** | **0.979** | **0.960** | **0.927** | **0.942** | **0.940** | **0.980** | **0.980** |
| Gemini-2.0-Flash (with image inputs) | 0.877 | 0.863 | 0.856 | 0.819 | 0.752 | 0.727 | 0.794 | 0.772 |
| Qwen2.5-VL-7B-Instruct | 0.613 | 0.545 | 0.612 | 0.544 | 0.611 | 0.543 | 0.613 | 0.545 |
| Phi-3.5-vision-instruct | 0.662 | 0.658 | 0.794 | 0.794 | 0.807 | 0.807 | 0.790 | 0.790 |
| **Language Models for Time Series** | | | | | | | | |
| ChatTime-7B-Chat | 0.321 | 0.319 | 0.279 | 0.276 | 0.261 | 0.257 | 0.301 | 0.300 |
| ChatTS-14B | **0.794** | **0.793** | **0.781** | **0.781** | **0.697** | **0.693** | **0.746** | **0.742** |

Table 7: Performance of Language and Vision-Language Models on SUSHI Dataset in the Recognition Setting using 4 Different Negative Sampling Techniques

Table 7 reports Recognition results on the SUSHI dataset, which features long, synthetic time series with well-structured seasonal and trend components. Performance is uniformly higher here than on other datasets, with GPT-4o (with image input) achieving near-ceiling F1 scores across all distractor types. ChatTS-14B performs particularly well in this setting, even surpassing many open-source LLMs, while smaller models such as LLaMA and Phi continue to underperform—highlighting the importance of both modality and model capacity when reasoning over long sequences.

| Model | S-Bert Embeddings with Cosine Similarity | | DTW Distance | | Longest Common Subsequence | |
|---|---|---|---|---|---|---|
| | Accuracy | F1 Score | Accuracy | F1 Score | Accuracy | F1 Score |
| **Language Models** | | | | | | |
| GPT-4o | 0.787 | 0.754 | 0.730 | 0.687 | 0.703 | 0.684 |
| GPT-4o (LLMTime) | **0.861** | **0.845** | **0.789** | **0.755** | **0.887** | **0.881** |
| Gemini-2.0-Flash | 0.720 | 0.694 | 0.688 | 0.633 | 0.662 | 0.614 |
| Llama-3.1-8B-Instruct | 0.514 | 0.367 | 0.506 | 0.364 | 0.508 | 0.362 |
| Qwen2.5-14B-Instruct-1M | 0.620 | 0.536 | 0.586 | 0.506 | 0.582 | 0.486 |
| Qwen2.5-7B-Instruct-1M | 0.612 | 0.529 | 0.573 | 0.492 | 0.570 | 0.470 |
| Phi-3.5-mini-instruct | 0.600 | 0.528 | 0.570 | 0.501 | 0.527 | 0.499 |
| **Vision-Language Models** | | | | | | |
| GPT-4o (with image inputs) | **0.924** | **0.922** | **0.875** | **0.873** | **0.930** | **0.912** |
| Gemini-2.0-Flash (with image inputs) | 0.822 | 0.812 | 0.780 | 0.750 | 0.735 | 0.731 |
| Qwen2.5-VL-7B-Instruct | 0.718 | 0.714 | 0.733 | 0.727 | 0.738 | 0.733 |
| Phi-3.5-vision-instruct | 0.680 | 0.673 | 0.682 | 0.679 | 0.637 | 0.615 |
| **Language Models for Time Series** | | | | | | |
| ChatTime-7B-Chat | 0.247 | 0.242 | 0.248 | 0.247 | 0.266 | 0.261 |
| ChatTS-14B | **0.803** | **0.779** | **0.774** | **0.771** | **0.833** | **0.832** |

Table 8: Performance of Various Language and Vision-Language Models on TaxoSynth Dataset in the Recognition Setting using 3 Different Negative Sampling Techniques

Table 8 presents Recognition results on the TaxoSynth dataset, which features synthetically generated sequences of varying lengths designed to reflect a taxonomy of distinct time series behaviors. Performance patterns largely mirror those seen in SUSHI, with GPT-4o (with image input) achieving the strongest scores across all distractor types. ChatTS-14B continues to perform competitively, while GPT-4o with LLMTime remains the strongest language-only model, reaffirming the value of instruction tuning and visual input on complex, structured sequence data.

| Model | S-Bert Embeddings with Cosine Similarity | | DTW Distance | | Euclidean Distance | | Longest Common Subsequence | |
|---|---|---|---|---|---|---|---|---|
| | Accuracy | F1 Score | Accuracy | F1 Score | Accuracy | F1 Score | Accuracy | F1 Score |
| **Language Models** | | | | | | | | |
| GPT-4o | 0.531 | 0.531 | 0.524 | 0.523 | 0.514 | 0.514 | 0.566 | 0.565 |
| GPT-4o (LLMTime) | **0.584** | **0.574** | **0.562** | **0.557** | **0.543** | **0.543** | **0.620** | **0.590** |
| Gemini-2.0-Flash | 0.505 | 0.501 | 0.529 | 0.528 | 0.518 | 0.517 | 0.499 | 0.497 |
| Llama-3.1-8B-Instruct | 0.310 | 0.261 | 0.259 | 0.195 | 0.242 | 0.183 | 0.249 | 0.182 |
| Qwen2.5-14B-Instruct-1M | 0.471 | 0.470 | 0.534 | 0.533 | 0.521 | 0.520 | 0.431 | 0.429 |
| Qwen2.5-7B-Instruct-1M | 0.474 | 0.465 | 0.429 | 0.414 | 0.438 | 0.422 | 0.290 | 0.265 |
| Phi-3.5-mini-instruct | 0.468 | 0.467 | 0.320 | 0.322 | 0.318 | 0.319 | 0.319 | 0.322 |
| **Vision-Language Models** | | | | | | | | |
| GPT-4o (with image inputs) | **0.631** | **0.630** | **0.669** | **0.669** | **0.657** | **0.657** | 0.541 | 0.541 |
| Gemini-2.0-Flash (with image inputs) | 0.608 | 0.592 | 0.650 | 0.607 | 0.604 | 0.600 | 0.501 | 0.500 |
| Qwen2.5-VL-7B-Instruct | 0.526 | 0.525 | 0.623 | 0.624 | 0.621 | 0.621 | 0.481 | 0.481 |
| Phi-3.5-vision-instruct | 0.589 | 0.588 | 0.612 | 0.613 | 0.603 | 0.604 | **0.577** | **0.559** |
| **Language Models for Time Series** | | | | | | | | |
| ChatTime-7B-Chat | 0.040 | 0.038 | 0.074 | 0.071 | 0.072 | 0.069 | 0.048 | 0.045 |
| ChatTS-14B | **0.492** | **0.492** | **0.412** | **0.409** | **0.393** | **0.389** | **0.501** | **0.499** |

Table 9: Performance of Various Language and Vision-Language Models on TRUCE-Stock Dataset in the Differentiation Setting using 4 Different Negative Sampling Techniques

Table 9 shows Differentiation task performance on the TRUCE-Stock dataset. As in the Recognition setting, this real-world dataset proves difficult across all model classes, with notably lower F1 scores. GPT-4o with LLMTime again leads among LLMs, and GPT-4o with image input remains strongest overall, though margins are narrower than in Recognition. ChatTS-14B continues to outperform most open-source LLMs, including its base LLM, reinforcing its strength in fine-grained comparison tasks under realistic conditions, even on the more difficult Differentiation setting.

| Model | S-Bert Embeddings with Cosine Similarity | | DTW Distance | | Euclidean Distance | | Longest Common Subsequence | |
|---|---|---|---|---|---|---|---|---|
| | Accuracy | F1 Score | Accuracy | F1 Score | Accuracy | F1 Score | Accuracy | F1 Score |
| **Language Models** | | | | | | | | |
| GPT-4o | 0.751 | 0.752 | 0.725 | 0.724 | 0.696 | 0.696 | 0.692 | 0.692 |
| GPT-4o (LLMTime) | **0.832** | **0.809** | **0.794** | **0.794** | **0.804** | **0.809** | **0.787** | **0.786** |
| Gemini-2.0-Flash | 0.745 | 0.741 | 0.629 | 0.625 | 0.592 | 0.591 | 0.690 | 0.665 |
| Llama-3.1-8B-Instruct | 0.342 | 0.284 | 0.275 | 0.217 | 0.267 | 0.210 | 0.257 | 0.200 |
| Qwen2.5-14B-Instruct-1M | 0.720 | 0.721 | 0.681 | 0.683 | 0.656 | 0.656 | 0.616 | 0.617 |
| Qwen2.5-7B-Instruct-1M | 0.612 | 0.607 | 0.470 | 0.463 | 0.445 | 0.433 | 0.417 | 0.402 |
| Phi-3.5-mini-instruct | 0.618 | 0.619 | 0.373 | 0.379 | 0.369 | 0.369 | 0.346 | 0.347 |
| **Vision-Language Models** | | | | | | | | |
| GPT-4o (with image inputs) | **0.801** | **0.801** | **0.808** | **0.807** | **0.769** | **0.769** | **0.729** | **0.728** |
| Gemini-2.0-Flash (with image inputs) | 0.789 | 0.786 | 0.745 | 0.739 | 0.727 | 0.725 | 0.704 | 0.700 |
| Qwen2.5-VL-7B-Instruct | 0.754 | 0.754 | 0.732 | 0.732 | 0.719 | 0.718 | 0.665 | 0.666 |
| Phi-3.5-vision-instruct | 0.802 | 0.802 | 0.708 | 0.709 | 0.694 | 0.693 | 0.657 | 0.658 |
| **Language Models for Time Series** | | | | | | | | |
| ChatTime-7B-Chat | 0.130 | 0.127 | 0.104 | 0.101 | 0.109 | 0.107 | 0.134 | 0.134 |
| ChatTS-14B | **0.654** | **0.652** | **0.593** | **0.588** | **0.601** | **0.598** | **0.711** | **0.708** |

Table 10: Performance of Various Language and Vision-Language Models on TRUCE-Synthetic Dataset in the Differentiation Setting using 4 Different Negative Sampling Techniques

Table 10 presents Differentiation performance on the TRUCE-Synthetic dataset. Overall scores are slightly lower than in the Recognition setting, reflecting the added complexity of multi-choice reasoning. GPT-4o with LLMTime and GPT-4o with image input lead across all sampling strategies, while ChatTS-14B again stands out among TSLMs, outperforming many open-source LLMs, including its base LLM. This pattern underscores the benefit of both vision-based inputs and domain-specific training for tasks requiring contrastive reasoning.

| Model | S-Bert Embeddings with Cosine Similarity | | DTW Distance | | Euclidean Distance | | Longest Common Subsequence | |
|---|---|---|---|---|---|---|---|---|
| | Accuracy | F1 Score | Accuracy | F1 Score | Accuracy | F1 Score | Accuracy | F1 Score |
| **Language Models** | | | | | | | | |
| GPT-4o | 0.802 | 0.803 | 0.646 | 0.648 | 0.583 | 0.584 | 0.916 | 0.916 |
| GPT-4o (LLMTime) | **0.850** | **0.823** | **0.673** | **0.670** | **0.615** | **0.614** | **0.975** | **0.959** |
| Gemini-2.0-Flash | 0.728 | 0.711 | 0.625 | 0.549 | 0.527 | 0.488 | 0.880 | 0.827 |
| Llama-3.1-8B-Instruct | 0.385 | 0.382 | 0.226 | 0.214 | 0.239 | 0.221 | 0.380 | 0.376 |
| Qwen2.5-14B-Instruct-1M | 0.740 | 0.738 | 0.446 | 0.431 | 0.365 | 0.351 | 0.792 | 0.792 |
| Qwen2.5-7B-Instruct-1M | 0.636 | 0.633 | 0.239 | 0.216 | 0.178 | 0.150 | 0.557 | 0.540 |
| Phi-3.5-mini-instruct | 0.643 | 0.648 | 0.444 | 0.443 | 0.344 | 0.344 | 0.716 | 0.721 |
| **Vision-Language Models** | | | | | | | | |
| GPT-4o (with image inputs) | **0.981** | **0.981** | **0.965** | **0.965** | **0.951** | **0.951** | **0.988** | **0.988** |
| Gemini-2.0-Flash (with image inputs) | 0.976 | 0.969 | 0.896 | 0.894 | 0.898 | 0.893 | 0.972 | 0.940 |
| Qwen2.5-VL-7B-Instruct | 0.974 | 0.974 | 0.871 | 0.871 | 0.851 | 0.851 | 0.968 | 0.968 |
| Phi-3.5-vision-instruct | 0.808 | 0.807 | 0.756 | 0.755 | 0.716 | 0.713 | 0.864 | 0.865 |
| **Language Models for Time Series** | | | | | | | | |
| ChatTime-7B-Chat | 0.287 | 0.285 | 0.235 | 0.231 | 0.261 | 0.260 | 0.281 | 0.283 |
| ChatTS-14B | **0.783** | **0.781** | **0.602** | **0.595** | **0.574** | **0.570** | **0.809** | **0.808** |

Table 11: Performance of Various Language and Vision-Language Models on SUSHI Dataset in the Differentiation Setting using 4 Different Negative Sampling Techniques

Table 11 summarizes Differentiation results on the SUSHI dataset. GPT-4o with image input achieves near-perfect F1 scores across all distractor types, followed closely by other VLMs, confirming the advantage of visual modality in structured settings. GPT-4o with LLMTime continues to lead among text-only models, while ChatTS-14B again proves competitive, performing on par with or better than many open-source LLMs, including its base LLM.

| Model | S-Bert Embeddings with Cosine Similarity | | DTW Distance | | Longest Common Subsequence | |
|---|---|---|---|---|---|---|
| | Accuracy | F1 Score | Accuracy | F1 Score | Accuracy | F1 Score |
| **Language Models** | | | | | | |
| GPT-4o | 0.716 | 0.716 | 0.709 | 0.710 | 0.655 | 0.655 |
| GPT-4o (LLMTime) | **0.794** | **0.781** | **0.752** | **0.747** | **0.696** | **0.687** |
| Gemini-2.0-Flash | 0.642 | 0.639 | 0.607 | 0.602 | 0.654 | 0.633 |
| Llama-3.1-8B-Instruct | 0.394 | 0.283 | 0.259 | 0.254 | 0.287 | 0.282 |
| Qwen2.5-14B-Instruct-1M | 0.668 | 0.667 | 0.527 | 0.520 | 0.658 | 0.658 |
| Qwen2.5-7B-Instruct-1M | 0.589 | 0.584 | 0.349 | 0.344 | 0.455 | 0.447 |
| Phi-3.5-mini-instruct | 0.596 | 0.593 | 0.495 | 0.497 | 0.524 | 0.528 |
| **Vision-Language Models** | | | | | | |
| GPT-4o (with image inputs) | **0.805** | **0.805** | **0.812** | **0.812** | **0.797** | **0.797** |
| Gemini-2.0-Flash (with image inputs) | 0.707 | 0.706 | 0.731 | 0.725 | 0.745 | 0.732 |
| Qwen2.5-VL-7B-Instruct | 0.767 | 0.763 | 0.759 | 0.753 | 0.748 | 0.748 |
| Phi-3.5-vision-instruct | 0.719 | 0.700 | 0.689 | 0.686 | 0.708 | 0.712 |
| **Language Models for Time Series** | | | | | | |
| ChatTime-7B-Chat | 0.175 | 0.172 | 0.150 | 0.147 | 0.176 | 0.176 |
| ChatTS-14B | **0.756** | **0.754** | **0.650** | **0.647** | **0.786** | **0.785** |

Table 12: Performance of Various Language and Vision-Language Models on TaxoSynth Dataset in the Differentiation Setting using 3 Different Negative Sampling Techniques

Table 12 reports Differentiation results on the TaxoSynth dataset. GPT-4o with LLMTime and GPT-4o with image input perform best overall, with consistent results across all sampling strategies. ChatTS-14B achieves strong performance, narrowing the gap with proprietary models and outperforming its base LLM, showing robustness even in longer, more complex multi-choice tasks.

These results offer fine-grained insight into model behavior and reinforce that our main findings are robust across contrastive construction strategies. (1) Across datasets, we find that vision-language models consistently outperform their text-only counterparts. (2) This performance gap is most evident in synthetic datasets like SUSHI and TRUCE-Synthetic, where models such as GPT-4o with image input achieve near-perfect F1 scores. (3) In contrast, the real-world TRUCE-Stock dataset presents a more substantial challenge, with significantly lower performance across all models, including proprietary LLMs and VLMs. (4) Recognition remains consistently easier than Differentiation, with a uniform drop in performance observed across models when switching tasks. (5) While vision-based models lead in most settings, strong time-series-specific language models like ChatTS-14B perform competitively in the Differentiation task, especially on datasets such as TaxoSynth where they outperform many open-source and some proprietary LLMs. (6) We further observe that annotation-based distractor construction yields the most consistent and stable results across tasks and models, with performance dropping more notably under time series-based contrast sets, particularly for smaller or less tuned LLMs like Phi and LLaMA. (7) Instruction tuning (as in GPT-4o + LLMTime) has a more pronounced effect on performance than model scale alone, and consistently improves robustness across both tasks. (8) Finally, model rankings remain remarkably stable across all datasets, contrastive sampling strategies, and task settings, underscoring that the observed trends are not artifacts of any specific evaluation construction. This consistency holds across synthetic and real-world datasets, across short and long time series, and across both simple and complex descriptions. The fact that model ordering is preserved despite this breadth of variation demonstrates the robustness and diagnostic reliability of the BEDTime.

*Note:* Euclidean distance was excluded for TaxoSynth due to its variable-length sequences, which prevent direct computation.

# E  IMPLEMENTATION DETAILS

Experiments are run through the OpenAI GPT-4o API and the Google Gemini API and model inference endpoints for Qwen, Phi, Llama, ChatTS and ChatTime models. Batching is used where possible to minimize API overhead. Inference is parallelized across NVIDIA A6000 GPUs for models requiring local deployment. Each experiment is repeated with all distractor sampling methods to ensure robustness of results.

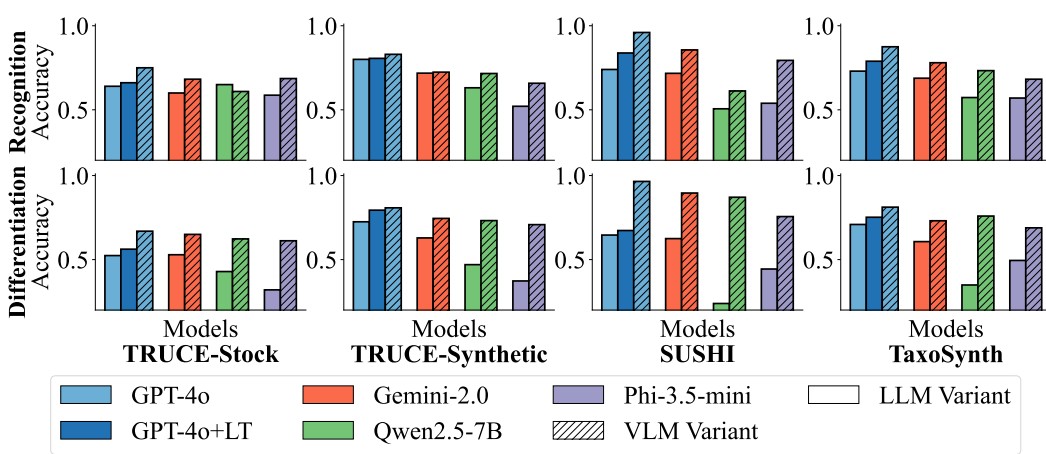

Figure 4: Accuracy of LLMs and VLMs on recognition and differentiation tasks across real-world (TRUCE-Stock) and synthetic (TRUCE-Synthetic, SUSHI, TaxoSynth) time series datasets. Negative samples for contrastive evaluation were generated using Dynamic Time Warping (DTW). The consistent performance gains of VLMs, especially on the differentiation task, highlight the importance of visual cues for robust time series analysis.

# F  WHY DOES CHATTIME PERFORM WORSE THAN RANDOM GUESSING ON RECOGNITION AND DIFFERENTIATION TASKS?

Upon analysis, we found that ChatTime often fails to follow the task instructions, which significantly affects its performance in these classification-based tasks. Specifically, rather than outputting a discrete label (e.g., "A/B/C/D" or "True/False"), ChatTime frequently returns a numeric value extracted directly from the input time series. Since our evaluation requires a well-formed answer to compare against the gold label, any non-conforming output is marked as incorrect, which brings down both accuracy and F1 considerably. To better understand this behavior, we tracked refusal rates (Refer to Table 1), the percentage of prompts where ChatTime did not produce a valid classification. As shown in Table below, these refusal rates are very high, especially in recognition tasks. We hypothesize that ChatTime's high error rates on instruction-following tasks like recognition and differentiation stem from its architectural and training design. While ChatTime incorporates instruction fine-tuning, its base LLM is LLaMA-2-7B-Base, a non-instruction tuned model that undergoes continuous pretraining on time series data before instruction tuning. The fine-tuning phase focuses on generation-based tasks such as forecasting and time series QA, but not on multiple-choice classification with strict formatting constraints. As a result, ChatTime often returns only numeric outputs rather than discrete classification labels (e.g., "A", or "True"), especially when the prompt requires strict instruction-following behavior. Table13 - Refusal rates for both Recognition and differentiation tasks across all four datasets, using DTW-based negative sampling.

| Task | Dataset | Refusal Rate (%) |
|---|---|---|
| **Recognition** | TRUCE-Stock | 83.4 |
| | TRUCE-Synthetic | 79.1 |
| | SUSHI | 61.6 |
| | TaxoSynth | 64.3 |
| **Differentiation** | TRUCE-Stock | 74.2 |
| | TRUCE-Synthetic | 67.0 |
| | SUSHI | 53.5 |
| | TaxoSynth | 58.7 |

Table 13: Refusal rates across recognition and differentiation tasks on different datasets.

# G   ADDITIONAL BASELINE METRICS

We ran additional Baseline comparison metrics against the generated descriptions by the top performer LLM, VLM and TSLM and the ground truth on SUSHI dataset:**Average Edit Distance:**GPT-4o Vision: 473.15, ChatTS: 426.22,GPT-4o: 434.68.**Average Token Overlap:** GPT-4o Vision: 0.071, ChatTS: 0.068, GPT-4o: 0.067.

The low token overlap and moderate edit distances suggest that exact matching metrics (such as BLEU or token overlap) may not fully capture quality in this task. These results support our choice to favor NLI-based entailment over string-based heuristics such as token overlap or edit distance.

Additionally, we recomputed the NLI metrics for the testing samples that include no numbers. This provides a result on samples where numeric alterations cannot be a problem. Our results are in Table 14 below, where "w/ #" denotes the dataset with all samples (N=1400), including those with numbers as in the paper, and "w/o #" denotes the dataset without numbers (N=1279). Here we see that there is minimal difference between the scores for datasets with and without numbers, suggesting numeric misalignment have minimal impact.

| Model | Bi-dir | Gen → GT | GT → Gen |
|---|---|---|---|
| GPT-4o-Vision (w/ #) | 14.41 | 47.65 | 15.29 |
| GPT-4o-Vision (w/o #) | 15.76 | 49.84 | 16.72 |
| GPT-4o (w/ #) | 2.94 | 9.71 | 6.18 |
| GPT-4o (w/o #) | 3.22 | 10.61 | 6.43 |
| ChatTS (w/ #) | 2.65 | 23.53 | 2.65 |
| ChatTS (w/o #) | 2.89 | 25.40 | 2.89 |

Table 14: DeBERTa NLI entailment percentages when ground truth numeric values are included (w/#) and excluded (w/o #).

# H   INTERPOLATION

Figure 5 supports the analysis in Section 4.4.1 by evaluating whether scaling interpolated values mitigates the tokenization-related degradation observed in LLMs. Performance improves for most models, except Qwen2.5-14B due to reasons previously discussed.

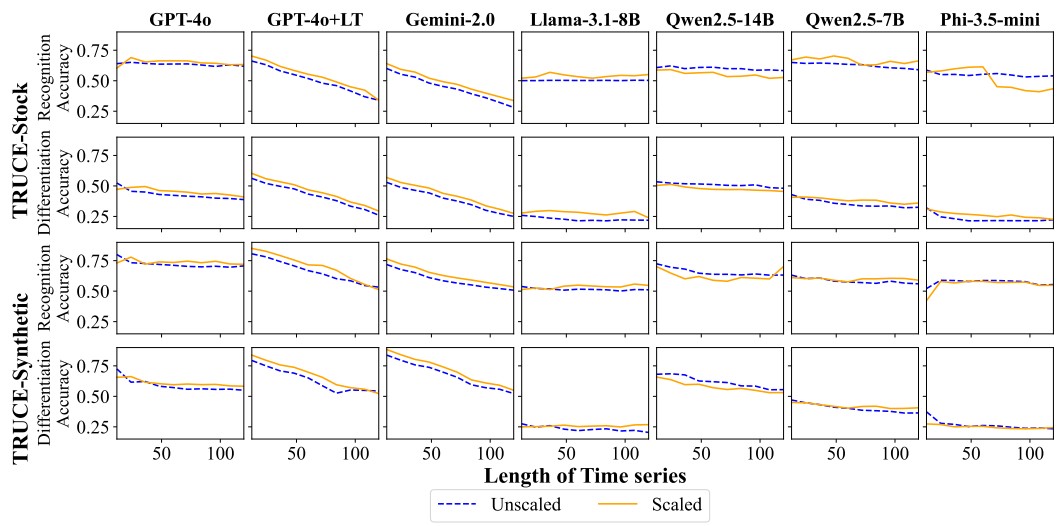

Figure 5: Scaling for different time series lengths

