# OpenReview forum: "BEDTime: A Unified Benchmark for Automatically Describing Time Series"
_ICLR.cc/2026/Conference — Submitted to ICLR 2026_

### Official Review · Reviewer_iBNc · 2025-10-22

**Soundness:** 3
**Presentation:** 3
**Contribution:** 3
**Rating:** 4
**Confidence:** 3

**Summary:**

This paper introduces BEDTime, a benchmark for evaluating language models' ability to recognize, differentiate, and generate natural language descriptions of univariate time series. The benchmark unifies four existing datasets (three synthetic, one real-world) and evaluates 13 models across three modalities: text-only LLMs, vision-language models (VLMs), and time-series language models (TSLMs). Key findings include: (1) VLMs substantially outperform other modalities, (2) dedicated TSLMs underperform despite being designed for similar tasks, and (3) all models show fragility under realistic perturbations like missing data and noise.

**Strengths:**

**S1. Addresses important evaluation gap**: Existing work evaluates LLMs, VLMs, and TSLMs in isolation using different datasets and metrics. This paper provides the first unified benchmark enabling direct comparison across modalities. The finding that VLMs outperform specialized TSLMs is surprising and valuable for the community.

**S2. Comprehensive experimental coverage**: The evaluation is exceptionally thorough. Tables 2-12 systematically present results across all conditions. This experimental rigor strengthens the empirical claims.

**S3. Valuable practical insights:** The finding that VLMs consistently outperform specialized TSLMs is surprising and actionable.

**S4. Clear presentation of results**: Tables 2-12 and Figures 2-5 systematically present results across conditions, making it easy to trace findings back to specific experimental settings.

**Weaknesses:**

**W1. Inadequate TSLM evaluation:** Only two TSLMs are evaluated: ChatTime and ChatTS. Appendix F reveals ChatTime has severe instruction-following failures with 61-83% refusal rates. This leaves only one functional TSLM. Drawing conclusions like "TSLMs severely underperform" from a single working model is unjustified. Recent surveys cite 10+ relevant TSLMs including Time-LLM, LLaTA, UniTime, and TEST. The omission of these models is a fundamental flaw. The paper's central claim about TSLM underperformance cannot be trusted without broader model coverage.

**W2. Unfair input representation across modalities:** VLMs receive matplotlib plots with axes, labels, and grids. LLMs receive raw comma-separated numbers. TSLMs get Python lists. These are not equivalent inputs. The matplotlib formatting provides structural cues (trend direction from slope, magnitude from visual height) that raw numbers lack. VLM superiority may reflect representation advantage rather than visual reasoning ability. The paper lacks ablations: What if VLMs received raw plots without annotations? What if LLMs received formatted tables? Without these controls, the fairness of comparison is questionable.

**W3. Over-reliance on synthetic data:** Three of four datasets are synthetic. The only real dataset (TRUCE-Stock) shows 20-30% lower accuracy than synthetic ones. The paper does not analyze why real data is harder or what properties cause difficulty. Claims about real-world applicability are weakened when 75% of evaluation uses synthetic data. The benchmark may not predict deployment performance.

**W4. NLI metric has unvalidated domain mismatch:** The generation task uses NLI models trained on general text to evaluate time series descriptions. Table 3 shows extremely low bi-directional entailment (2.65-14.41%). This indicates either poor generation quality or metric failure. The paper acknowledges domain mismatch but provides no validation that NLI scores correlate with human judgments. Without this validation, generation results may be unreliable.

**W5. Limited human evaluation:** Only 340 samples (24% of SUSHI) are human-evaluated by "research group members." This introduces potential bias. Standard practice uses external annotators on larger samples (1000+). The small sample size limits statistical power for comparing 13 models across 6 criteria.

**W6. Oversimplified negative sampling:** Distractors are selected as "most dissimilar" time series via DTW/Euclidean distance. This creates easy discrimination. Real errors involve subtle confusion between similar patterns. Table 7 shows 96-98% accuracy on SUSHI, suggesting ceiling effects. The benchmark may overestimate model robustness.

**W7. Critical results buried in Appendix:** Chain-of-Thought prompting shows 10-15% accuracy improvements (Figure 3) but appears only in Appendix E.2. ChatTime's 83% refusal rate appears only in Appendix F. These are major findings that should be in the main text.

**W8. Missing diagnostic analyses:** No class-stratified results (trends vs. seasonality vs. anomalies). No length-based breakdown despite variable sequence lengths. No error analysis of what mistakes models make. These missing analyses limit diagnostic value.

**Questions:**

Q1. Why is real data performance so much lower? TRUCE-Stock shows accuracy much lower than synthetic data. What properties of real series cause difficulty? Does synthetic data dominance limit real-world validity?

---

> ### Author Response · Authors · 2025-11-25
> **Rebuttal Response Part-1**
>
> We thank the reviewer for their thoughtful and constructive feedback. We are encouraged by their recognition of the important evaluation gap BEDTime addresses by providing the first unified benchmark for comparing LLMs, VLMs, and TSLMs. We also appreciate their emphasis on the surprising and valuable finding that VLMs outperform specialized TSLMs, as well as their acknowledgement of the actionable practical insights enabled by these results and the clear, systematic presentation of our tables and figures. We further appreciate their recognition of our exceptionally thorough experimental coverage across all datasets, models, and perturbation settings. We address the reviewer’s concerns and comments below:
>
> **W1.** TSLM evaluation is too limited—only one functional model is effectively tested—so broad claims about TSLM underperformance are not well supported, especially given many omitted models like Time-LLM, LLaTA, UniTime, and TEST.
>
> We agree that many TSLMs exist; however, most cannot operate on the time series description tasks defined by BEDTime. To clarify this, we carefully examined all mentioned models and several others. The table below summarizes why each cannot perform recognition, differentiation, or natural-language description generation defined in BEDTime.
>
> |**Model**|**Takes TS + Text?**|**Main Abilities**|**Natural Language Generation?**|**Why Not Usable in BEDTime**|
> |-|-|-|-|-|
> |**TEST** [1]|**Yes**|Classification, forecasting|**No**|Never trained to interpret text about TS or produce descriptions; architecture routes TS through LLM for numeric outputs only, so not suited for description recognition, differentiation, or generation tasks.|
> |**Time-VLM** [2]|**Yes** |Forecasting|**No**|Text is only contextual input; the model fuses temporal/visual/text embeddings solely to improve numeric prediction; cannot produce or recognize or differentiate time series descriptions.|
> |**Time-LLM** [3]|**Yes**|Forecasting|**No**|LLM backbone is reprogrammed for numeric forecasting with a projection head; no TS–text semantic matching or decoding; cannot generate or recognize or differentiate descriptions.|
> | **CALF** (previously LLaTa) [4]|**Yes**|Forecasting|**No**|All outputs are numeric hence cannot generate or identify descriptions.|
> |**UniTime** [5]|**Yes**|Forecasting|**No**|Language tokens serve as domain identifiers, not descriptors; model predicts future numeric sequence values only; lacks any description generation component.|
> |**UNITS** [6]|**No**|Forecasting, classification, imputation, anomaly detection|**No**|The model is TS-only with no text interface; cannot take in text descriptions, compare TS with text, or generate text, hence fails to perform all  BEDTime tasks.|
> |**CLaSP** [7]|**Yes**|Contrastive alignment, retrieval|**Yes** (retrieves but does not generate)|Learns embeddings for nearest-neighbor text retrieval—not generation; cannot evaluate correctness of text or create new descriptions; only retrieves closest caption hence cannot identify or generate time series descriptions.|
>
> In addition, we examined two recent TSLMs designed explicitly with time series and text functionalities [8,9], but neither releases code, pretrained weights, or the large proprietary and synthetic time series and text datasets required to train or reproduce the models, making them impossible to evaluate on BEDTime.
>
>
>
> **W2.** Input formats across modalities are inconsistent such as VLMs get richly formatted plots while LLMs and TSLMs get raw numbers, thereby making the comparison unfair.
>
> We’d like to clarify that VLMs do not actually receive any information that is not already available in the numeric vectors provided to LLMs and TSLMs. This is because the plots are rendered with a neutral matplotlib template that exposes only what the raw values already contain. The x-axis is simply the index timestep (datasets include no timestamps), and the y-axis is just the numeric scale of the supplied values. No labels, titles, grids, color cues, metadata, annotations, or domain hints are added. We will make this clearer in the paper where we describe the plot generation.

---

> ### Author Response · Authors · 2025-11-25
> **Rebuttal Response Part-2**
>
> **W3/Q1.** BEDTime relies on three synthetic datasets, and the one real dataset yields substantially lower accuracy, but the paper doesn’t analyze why real data is harder or how this affects real-world validity.
>
> We agree that TRUCE-Stock yields lower accuracy than the synthetic datasets. This pattern is expected as TRUCE-Stock contains real, noisy financial signals paired with diverse human-written captions, whereas the synthetic datasets contain cleaner temporal dynamics and structured descriptions. This difference naturally increases task difficulty on TRUCE-Stock and mirrors the degradation we explicitly analyze in our real-world robustness tests (Figure 2), where models drop similarly under interpolation, noise, missingness, and scaling. These findings indicate that the gap arises from signal complexity and linguistic variability.
>
> To further explore this, we have added a new real-world, clinically-grounded dataset containing NICU heart-rate time series signal data, patient metadata, and validated clinical descriptions [10]. We describe the dataset and experiments below, which will also be added to the paper.
> The added NICU dataset includes 36,679 public real heart-rate recordings from infants treated in a NICU from 2012–2016. Each 10-minute (300-step, 2-second sampled) sequence is paired with:
>
> - Natural language clinical descriptions derived from clinical bradycardia events and variability annotations that were clinically validated for semantic diversity and correctness.
>
> - Patient metadata, that includes demographics, delivery mode, Apgar scores, comorbidities, gestational age, birth weight, maternal age, race, etc.
>
> We evaluate whether models can identify real NICU heart-rate signals’ clinical descriptions through two tasks:
>
> - Description Recognition (True/False): Models judge whether a candidate description accurately reflects the heart rate time series signal. Here:
> The distractors come are chosen as the most different signal based on via DTW-distance. Distractors selection is constrained as to make sure that they always come from patients with opposite clinical outcomes.
>
> - Description Differentiation (Multiple Choice): Models select the correct description among four real clinical descriptions (1 correct and 3 distractors chosen in the same constraints as above).
>
> Both tasks are repeated with and without patient metadata, requiring models to integrate physiological patterns with etiological factors rather than relying only on visual cues. The table below shows the results for these experiments, which all align with our existing findings on the synthetic data. Bolstering our previous BEDTime results, VLMs lead across settings, with GPT-4o-Vision at the top. GPT-4o takes the lead amongst LLMs and benefits significantly with BPE-based LLMTime sequence inputs. As seen previously, TSLMs beat their base LLM baselines with ChatTS performing better than all LLMs of comparable size across all 4 settings. The introduction of Patient Metadata shows an interesting pattern: While there is an overall increase in model performance when patient metadata is introduced. Larger models(>14B) show greater improvement over lower parameter models (<7B) in this setting across both tasks.
>
> |Models|Recognition(TS)|Recognition(TS+Metadata)|Avg. Recognition Rank| Differentiation(TS)|Differentiation(TS+Metadata)|Avg. Differentiation Rank|Overall Rank|
> |-|-|-|-|-|-|-|-|
> |GPT-4o|0.567|0.640|2|0.944|0.971|2|2|
> |GPT-4o+LT|0.569|0.645|1|0.948|0.979|1|1|
> |Gemini-2.0|0.557|0.638|3|0.938|0.969|3|3|
> |Llama-3.1-8B|0.458|0.447|7|0.689|0.678|6.5|6.75|
> |Qwen2.5-14B|0.540|0.612|4|0.814|0.852|4|4|
> |Qwen2.5-7B|0.530|0.520|5|0.782|0.698|5|5|
> |Phi-3.5-mini|0.506|0.518|6|0.765|0.667|6.5|6.25|
> |GPT-4o-Vision|0.604|0.667|1|0.957|0.983|1|1|
> |Gemini-2.0-Vision|0.597|0.613|2|0.943|0.976|2|2|
> |Qwen2.5-VL-7B|0.544|0.553|3|0.793|0.721|3|3|
> |Phi-3.5-Vision|0.529|0.529|4|0.771|0.714|4|4|
> |ChatTime-7B|0.543|0.562|2|0.727|0.711|2|2|
> |ChatTS-14B|0.545|0.587|1|0.821|0.776|1|1|
>
> These experiments show that while adding and exploring new real-world data is valuable, it does not alter our conclusions and BEDTime remains to be representative of model behavior on external and out-of-distribution real-world data. We will incorporate this NICU study as an additional real-world case of descriptive and etiological time-series reasoning.

---

> ### Author Response · Authors · 2025-11-25
> **Rebuttal Response Part-3**
>
> **W4/W5.** The reliability of the Generation evaluation is unclear: the NLI metric is unvalidated for time-series text and extremely low entailment scores suggest domain mismatch, while the human evaluation is limited to 340 samples, raising concerns about the robustness of the generation results.
>
> We understand that unbiased evaluation of open-ended generation for time series descriptions is challenging. For this reason, our evaluation does not rely solely on NLI or solely on human annotation, but on a mix of both, where human evaluation (based on six criteria from prior literature) serves as the primary measure and DeBERTa NLI is used only as an auxiliary signal. To address domain mismatch, we compute NLI in two modes: (1) with numeric descriptions (“w/#”) and (2) with numeric descriptions removed (“w/o #”) to prevent absolute numeric values from influencing entailment. All three evaluation modes—w/# NLI, w/o # NLI, and human ratings—produce the exact same ranking on SUSHI, shown below:
>
> |Model|Average w/# NLI Rank|Average w/o # NLI Rank|Average Human Rater Rank|Overall Rank|
> |-|-|-|-|-|
> |GPT-4o-Vision|1.00|1.00|1.00|1.00|
> |GPT-4o|2.33|2.33|2.33|2.33|
> |ChatTS-14B|2.67|2.67|2.17|2.50|
>
> While raw DeBERTa entailment percentages are low, this is expected and not evidence of poor generation, because DeBERTa is extremely sensitive to lexical and syntactic variation. To further address this domain mismatch, we additionally evaluate generation using a GPT-5 as an LLM judge, which is known to perform far better on semantic alignment and paraphrase recognition than DeBERTa. In this setup, for each pair consisting of the ground-truth description and the model-generated description, we explicitly prompt GPT-5 to decide whether the generation entails the ground truth, whether the ground truth entails the generation, and if both conditions hold simultaneously we treat it as bi-directional entailment. This yields the following entailment-style scores, which again preserve the exact same model ranking:
>
> |Model|w/# Bi-dir %	|w/# Gen→GT %|w/# GT→Gen %|w/# Average Rank|
> |-|-|-|-|-|
> |GPT-4o-Vision |47.64|71.43|51.86|1|
> |GPT-4o|37.93|58.14|43.43|2|
> |ChatTS-14B|20.93|46.71|34.93|3|
>
> This demonstrates that all our generation evaluation methods yield stable, unbiased results that are fully robust to both domain mismatch and human-evaluation scale.
>
> **W6.** Negative sampling is oversimplified by using only the “most dissimilar” series making the task too easy, leading to ceiling effects and potentially overstating model robustness.
>
> We clarify that BEDTime is designed to establish a clear baseline for the time series description, where models must reliably reject clearly incorrect descriptions before tackling more challenging distractors. As part of this baseline, we evaluate models under four  negative-sampling strategies to test whether performance remains consistent across multiple constructions of the negative set. Importantly, our results show that the current setup is **far from trivial** where even with maximally dissimilar distractors, model performance drops markedly, especially under real-world perturbations such as noise, missingness, and interpolation. These declines suggest that models still struggle with basic discriminative reasoning making ceiling effects are not yet a concern. However, we agree that exploring harder negative sampling strategies is a valuable future direction and will add this to the limitations and future works section.
>
> **W7.** Important findings of accuracy gains from Chain-of-Thought prompting and ChatTime’s high refusal rate are not highlighted in the main text.
>
> We agree that these findings are noteworthy and can be made more visible. In the camera-ready version, we will highlight the accuracy gains from Chain-of-Thought prompting and ChatTime’s high refusal rate directly in the main text including the abstract, Introduction and the very beginning of the experimental discussions, so that readers can immediately understand their significance.
>
> **W8.** Missing diagnostic analyses of class-stratified performance, breakdown by sequence length, and error analysis, limiting insight into model failure cases.
>
> We’d like to clarify that BEDTime already includes several relevant diagnostic analyses. We directly examine sequence-length effects through our interpolation and scaled-interpolation robustness tests (Figures 2 and 5), which evaluate how models behave as the effective sequence length increases; these results show clear degradation for text-only LLMs and more stable performance for TSLMs. We also report failure patterns across multiple robustness experiments namely missingness, noise, amplitude scaling (Figure 2) as well as ChatTime’s refusal-rate analysis (Appendix F). Together, these analyses highlight where and how models fail. We will update the main text to more clearly highlight these analyses for readers.

---

> ### Author Response · Authors · 2025-11-25
> **Rebuttal Response Part-4**
>
> REFERENCE:
>
> [1]Sun et al., “TEST: Text Prototype Aligned Embedding to Activate LLM’s Ability for Time Series”, ICLR 2024.
>
> [2]Zhong et al., “Time-VLM: Exploring Multimodal Vision-Language Models for Augmented Time Series Forecasting”, ICML 2025.
>
> [3]Jin et al., “Time-LLM: Time Series Forecasting by Reprogramming Large Language Models”, ICLR 2024.
>
> [4] Liu et al., “CALF: Aligning LLMs for Time Series Forecasting via Cross-Modal Fine-Tuning”, AAAI 2025.
>
> [5]Liu et al., “UniTime: A Language-Empowered Unified Model for Cross-Domain Time Series Forecasting”, WWW 2024.
>
> [6]Gao et al., “UniTS: A Unified Time Series Model for Multi-Domain, Multi-Task Learning”, NeurIPS 2024.
>
> [7]Ito et al., “CLaSP: Learning Concepts for Time-Series Signals from Natural Language Supervision”, arXiv 2025.
>
> [8]Trabelsi et al., “Time Series Language Model for Descriptive Caption Generation”, arXiv 2025.
>
> [9]Chow et al., “Towards Time-Series Reasoning with LLMs”, NeurIPS 2024 (Time Series in the Age of Large Models Workshop).
>
> [10]Qiu et al., 2025. "Instruction-based Time Series Editing".

---

> > ### Comment · Reviewer_iBNc · 2025-11-27
> >
> > Thanks for the authors' detailed rebuttal. I would like to raise my rating.

---

### Official Review · Reviewer_za99 · 2025-10-23

**Soundness:** 3
**Presentation:** 3
**Contribution:** 3
**Rating:** 4
**Confidence:** 5

**Summary:**

In this paper, the authors proposed a new time series understanding benchmark datasets to evaluate the ability of LLM/VLM/TSLM to generate language descriptions of time series. By curating existing datasets, the authors create 3 tasks for evaluation: description recognition, description differentiation and open generation. Extensive experiments show the limitations of current LLM/VLM/TSLM.

**Strengths:**

S1: The paper proposed a new time series understanding datasets for the task of describing time series.
S2: Extensive evaluations of SOTA LLM/VLM/TSLM are performed.

**Weaknesses:**

W1: The paper did not talk much about how they ensure the quality of the generated benchmarks. For instance, the authors make sampling and select the most dissimilar option as the incorrect description. Human evaluation should be included to ensure the quality.
W2: Most of the included datasets are from existing work and they are mostly synthetic and small. For instance, the TRUCE dataset only contains 12 points. The largest one only contains 2048 points.

**Questions:**

Q1: Why not include more datasets into the benchmark. For instance, the one used in timeseriesexam and ChatTS.
Q2: The performance of VLM will be strongly affected by the resolution of the plotting of the time series. However, this part is not evaluated.
Q3: It seems that the questions are mostly from the caption of the time series data. However, the timeseries description task are frequently asked for local details. Can the proposed benchmark cover these aspects?

---

> ### Author Response · Authors · 2025-11-25
> **Rebuttal Response Part-1**
>
> We thank the reviewer for their thoughtful and constructive feedback. We are encouraged by their recognition of several core contributions of our work, including our introduction of a novel benchmark tailored to time-series description; our curation of four datasets into three clear evaluation tasks: recognition, differentiation, and open generation, thereby providing a comprehensive evaluation framework; and the breadth of our empirical analysis of state-of-the-art LLMs, VLMs, and TSLMs, which highlights important and previously unreported limitations across three input modalities. We address the reviewer’s concerns and comments below:
>
> **W1.** The paper doesn’t explain how data quality is ensured specifically for sampled negative descriptions.
>
> We thank the reviewer for their thoughtful comment. We take this opportunity to clarify that we ensure data curation quality of BEDTime by relying exclusively on previously peer-reviewed, or publicly released time-series–text datasets with established curation procedures.  BEDTime consists of four datasets from three independent sources: TRUCE-Stock and TRUCE-Synthetic, SUSHI , and TaxoSynth. We will update our dataset selection descriptions in Appendix A to clarify that their quality has been ensured by their parent sources as follows:
>
> - TRUCE-Stock descriptions are crowd-sourced on MTurk with strict annotator filters, with annotators being shown plots instead of tables to focus on patterns, hiding stock names to reduce bias, limiting captions to ≤9 words so each describes a single pattern, and having 3 annotators per item, manual inspection of annotators and removal of low-quality descriptions.
>
> - TRUCE-Synthetic also uses controlled, formula-based generation of patterns with light noise, ensuring clean ground-truth alignment between each series and its descriptions.
>
> - SUSHI contains fully synthetic signals generated from predefined functional classes; with text descriptions drawn from a pre-registered template list corresponding to these functions and then manually refined and visually verified by the dataset creators to ensure accuracy and consistency.
>
> - TaxoSynth is a synthetic dataset built from a clear, rigorous, and predefined taxonomy of time-series behaviors. Each pattern is deliberately prepared using parameterized functional generators and dataset-level isolation (one pattern per dataset). Post signal generation, an extra layer of quality control is added through checks of parameter bounds, which help ensure that every series genuinely reflects the characteristics it’s meant to represent.
>
> We also clarify that we ensure negative-sample quality adopting a rigorously controlled sampling process:
>
> - Distractors/Negatives are drawn only from real captions in the source/parent datasets and never synthesized or edited. This guarantees that every negative example remains within the same description distribution as its parent dataset.
>
> - We select negatives using four complementary dissimilarity strategies that jointly capture semantic, shape-based, and structural differences between time series and captions. We use Sentence-BERT embeddings with cosine-similarity, Dynamic Time Warping (DTW) distance, Euclidean distance, and Longest Common Subsequence (LCSS) distance. Using multiple metrics makes sure that our results and conclusions are robust.
>
> - Moreover, for each target pair, we choose the most dissimilar example under the selected metric. This ensures that each distractor represents a clearly different time-series pattern.
>
> - Furthermore, for datasets with class or subclass labels (SUSHI, TaxoSynth), we further constrain negatives to come strictly from different classes. This prevents accidental class/subclass overlap even when raw distance metrics might otherwise surface structurally similar signals from within the same category.
>
> Lastly, the entire negative sampling pipeline is deterministic and fully reproducible. All sampling code, including metric implementations, is provided to ensure complete transparency and reproducibility.
> For future clarity of the readers we will make an explicit note of these controls in the Dataset Description section in the Appendix of our manuscript.

---

> ### Author Response · Authors · 2025-11-25
> **Rebuttal Response Part-2a**
>
> **W2/Q1.**  The datasets are mostly small or synthetic, raising concerns about how well the benchmark generalizes to real-world settings and  would benefit from including additional datasets, such as TimeSeriesExam or ChatTS.
>
> We thank the reviewer for this thoughtful suggestion and fully agree that expanding BEDTime with additional datasets is a valuable direction. In fact, we have actively worked to extend the benchmark, and have already added a new real-world clinical dataset to the evaluation.
>
> However, although we agree with the spirit of the suggestion, we do not believe that TimeSeriesExam or the datasets used to train/evaluate ChatTS are appropriate candidates for inclusion in BEDTime. Below, we explain in detail why these datasets are structurally incompatible with data required for BEDTime’s three tasks and goal of evaluating a language models' ability to visually describe a time series.
>
> TimeSeriesExam is fundamentally incompatible with BEDTime because its data lacks the structure required for our evaluation. BEDTime’s tasks all require paired <time series, natural-language description> entries, while TimeSeriesExam contains only template-driven multiple-choice questions over synthetic pattern primitives. It evaluates conceptual time series reasoning like stationarity, anomaly type, similarity, and Granger-style causal structure and does not have any component to evaluate time series description identification or generation. The dataset includes no natural-language captions describing visual properties of time series (such as trend, seasonality, shifts, or noise), and its format cannot be transformed into BEDTime’s captioning tasks without manually creating new descriptions for every synthetic series.
>
> The datasets used in ChatTS (attribute-based generator corpora, TSEvol-generated Q&A, and evaluation sets A/B/MCQ2) are built for synthetic attribute alignment, correlation detection, and causal reasoning over primarily multivariate time series, making them linguistically and structurally mismatched with what BEDTime measures. These datasets contain no natural-language descriptions of visual patterns; instead, they include attribute-grounded explanations tied to internal generator metadata. This differs fundamentally from the crowd-written, template-written, and taxonomy-based descriptions in BEDTime, which reflect how humans visually describe univariate time series. Moreover, because ChatTS centers on multivariate, attribute-pooled synthetic data, it is inherently misaligned with BEDTime’s univariate descriptive focus.

---

> ### Author Response · Authors · 2025-11-25
> **Rebuttal Response Part-2b**
>
> To directly address the reviewer’s broader concern, we have added a new real-world, clinically-grounded dataset containing NICU heart-rate time series signal data, patient metadata, and validated clinical descriptions[1]. We describe the dataset and experiments below, which will also be added to the paper. The added NICU dataset includes 36,679 public real heart-rate recordings from infants treated in a NICU from 2012–2016. Each 10-minute (300-step, 2-second sampled) sequence is paired with:
>
> - Natural language clinical descriptions derived from clinical bradycardia events and variability annotations that were clinically validated for semantic diversity and correctness.
>
> - Patient metadata, that includes demographics, delivery mode, Apgar scores, comorbidities, gestational age, birth weight, maternal age, race, etc.
>
> We evaluate whether models can identify real NICU heart-rate signals’ clinical descriptions through two tasks:
>
> - Description Recognition (True/False): Models judge whether a candidate description accurately reflects the heart rate time series signal. Here:
> The distractors come are chosen as the most different signal based on via DTW-distance. Distractors selection is constrained as to make sure that they always come from patients with opposite clinical outcomes.
>
> - Description Differentiation (Multiple Choice): Models select the correct description among four real clinical descriptions (1 correct and 3 distractors chosen in the same constraints as above).
>
> Both tasks are repeated with and without patient metadata, requiring models to integrate physiological patterns with etiological factors rather than relying only on visual cues. The table below shows the results for these experiments, which all align with our existing findings on the synthetic data. Bolstering our previous BEDTime results, VLMs lead across settings, with GPT-4o-Vision at the top. GPT-4o takes the lead amongst LLMs and benefits significantly with BPE-based LLMTime sequence inputs. As seen previously, TSLMs beat their base LLM baselines with ChatTS performing better than all LLMs of comparable size across all 4 settings. The introduction of Patient Metadata shows an interesting pattern: While there is an overall increase in model performance when patient metadata is introduced. Larger models(>14B) show greater improvement over lower parameter models (<7B) in this setting across both tasks.
>
> |Models|Recognition(TS)|Recognition(TS+Metadata)|Avg. Recognition Rank| Differentiation(TS)|Differentiation(TS+Metadata)|Avg. Differentiation Rank|Overall Rank|
> |-|-|-|-|-|-|-|-|
> |GPT-4o|0.567|0.640|2|0.944|0.971|2|2|
> |GPT-4o+LT|0.569|0.645|1|0.948|0.979|1|1|
> |Gemini-2.0|0.557|0.638|3|0.938|0.969|3|3|
> |Llama-3.1-8B|0.458|0.447|7|0.689|0.678|6.5|6.75|
> |Qwen2.5-14B|0.540|0.612|4|0.814|0.852|4|4|
> |Qwen2.5-7B|0.530|0.520|5|0.782|0.698|5|5|
> |Phi-3.5-mini|0.506|0.518|6|0.765|0.667|6.5|6.25|
> |GPT-4o-Vision|0.604|0.667|1|0.957|0.983|1|1|
> |Gemini-2.0-Vision|0.597|0.613|2|0.943|0.976|2|2|
> |Qwen2.5-VL-7B|0.544|0.553|3|0.793|0.721|3|3|
> |Phi-3.5-Vision|0.529|0.529|4|0.771|0.714|4|4|
> |ChatTime-7B|0.543|0.562|2|0.727|0.711|2|2|
> |ChatTS-14B|0.545|0.587|1|0.821|0.776|1|1|
>
> From this we can conclude that while addition of new datasets is beneficial to BEDTime, the benchmark with its 4 datasets is representative of model performance on external/out-of-distribution data and our conclusions made in the paper holds. We will update our main manuscript to introduce this as a case-study on out of distribution data with etiological reasoning.

---

> ### Author Response · Authors · 2025-11-25
> **Rebuttal Response Part-3**
>
> **Q2.** VLM performance likely depends heavily on plot resolution, but this factor isn’t evaluated.
>
> We thank the reviewer for their thoughtful question. In order to address the reviewer’s concern regarding the effect of plotting resolution on VLM performance, we extend our robustness analysis in Section 4.2.3 to add a controlled pixelation study that isolates visual fidelity as the sole varying factor. In BEDTime, VLMs process matplotlib-rendered plots rather than raw numeric sequences; thus, understanding their sensitivity to image quality is essential for interpreting their performance in Description Recognition and Differentiation tasks. To evaluate this, we use the SUSHI dataset using DTW Distance for distractor selection and systematically degrade each plot’s resolution while keeping the underlying time series fixed, following a standard downscale–upscale protocol. Starting from the original plot, we downsample the image by factors (f) of {2,4,8,16,32} and then upsample it back using nearest-neighbor interpolation respectively, giving rise to six interpretable fidelity levels: Pristine (×1), Mild (×2), Moderate (×4), Heavy (×8), Extreme (×16), and Catastrophic (×32). This alters only visual quality but not the time series itself ensuring that any performance change reflects  fragility of the vision encoder to image quality, not variation in the core task. The Tables below reports accuracy across all six pixelation levels for each VLM for the sushi dataset for Recognition and Differentiation tasks.
>
> **Table 1 (a)** - Recognition
>
> |Model|Pristine (×1)|Mild (×2)|Moderate (×4)|Heavy (×8)|Extreme (×16)|Catastrophic (×32)|
> |-|-|-|-|-|-|-|
> |GPT-4o-Vision|0.96|0.82|0.78|0.66|0.54|0.48|
> |Gemini-2.0-Vision|0.86|0.81|0.71|0.54|0.51|0.38|
> |Qwen2.5-VL-7B|0.61|0.57|0.46|0.44|0.32|0.28|
> |Phi-3.5-Vision|0.79|0.77|0.61|0.57|0.45|0.32|
>
> **Table 1 (b)** - Differentiation
>
> |Model|Pristine (×1)|Mild (×2)|Moderate (×4)|Heavy (×8)|Extreme (×16)|Catastrophic (×32)|
> |-|-|-|-|-|-|-|
> |GPT-4o-Vision|0.97|0.92|0.88|0.72|0.63|0.51|
> |Gemini-2.0-Vision|0.90|0.87|0.74|0.64|0.57|0.48|
> |Qwen2.5-VL-7B|0.87|0.81|0.75|0.68|0.56|0.46|
> |Phi-3.5-Vision|0.76|0.71|0.68|0.59|0.51|0.44|
>
> Across both Recognition and Differentiation tasks, we observe a clear, monotonic degradation in VLM accuracy as pixelation severity increases. GPT-4o-Vision is the most robust model: although its accuracy declines below random baseline under heavy corruption, it maintains relatively high performance even at ×8 and ×16, suggesting strong tolerance to loss of fine-grained detail. Gemini-2.0-Vision follows a similar but slightly weaker pattern, showing faster degradation under heavier corruption. Qwen2.5-VL-7B and Phi-3.5-Vision exhibit substantially lower robustness. Qwen2.5-VL-7B drops sharply under moderate pixelation (×4). Phi-3.5-Vision performs better than Qwen2.5-VL-7B at low corruption but deteriorates more rapidly at extreme levels. In conclusion, VLMs are sensitive to plot resolution, but the degree of sensitivity varies widely across architectures and model sizes
>
> **Q3.** BEDTime appears to focus on high-level captions. Does it adequately capture local, fine-grained time-series description task performance?
>
> We thank the reviewer for this thoughtful observation. Our benchmark intentionally focuses on global visual descriptions, and we will make this clearer in the introduction of our manuscript to adjust expectations for readers. Our datasets indeed emphasize overall trends, consistent with prior work on time series description. This is also aligned with our Abstraction criterion in the generation evaluations, which explicitly states that descriptions should summarize overall patterns rather than step-by-step fluctuations. We agree this means BEDTime is not intended to capture fine-grained, timestamp-level descriptions. Producing such localized annotations would require datasets that include precise timing references, and to our knowledge, no such datasets currently exist. We will note in the paper that this is a limitation and an important direction for future work.
>
> REFERENCE:
>
> [1] Qiu et al., 2025. "Instruction-based Time Series Editing".

---

> > ### Comment · Reviewer_za99 · 2025-11-26
> >
> > Thanks for the authors' rebuttal, which addressed most of my concerns. I appreciated that the authors plan to add more controlled datasets to bedtime. Therefore, I would like to raise my overall score to 6.

---

### Official Review · Reviewer_KZgR · 2025-10-31

**Soundness:** 2
**Presentation:** 2
**Contribution:** 2
**Rating:** 4
**Confidence:** 4

**Summary:**

This paper introduces BEDTime, a unified benchmark designed to evaluate models' capabilities in describing time series data using natural language. The benchmark formalizes three tasks: (1) **Recognition** - true/false question-answering to assess whether a given statement about a time series is correct; (2) **Differentiation** - multiple-choice question-answering to select the most appropriate description; and (3) **Generation** - open-ended natural language description generation. The authors consolidate four recent datasets to enable direct model comparisons and evaluate 13 state-of-the-art models, including language-only models, vision-language models, and time-series language models.

The main findings indicate that language-only models perform poorly, vision-language models show promise, and pre-trained multimodal time-series language models outperform large language models but still have significant room for improvement. All methods demonstrate fragility in robustness tests. While the paper addresses an important need for standardized evaluation in time-series text multimodal understanding, several critical limitations regarding literature coverage, experimental design, and benchmark comprehensiveness prevent it from being a complete contribution to the field.

**Strengths:**

* **Well-defined task framework**: The formalization of three distinct tasks (Recognition, Differentiation, Generation) provides a clear structure for evaluating different aspects of time-series description capabilities. This taxonomy is useful for understanding model strengths and weaknesses across different reasoning types.
* **Comprehensive model evaluation**: The evaluation of 13 diverse models across different categories (language-only, vision-language, time-series language models) offers valuable insights into the current state of the field and reveals interesting patterns about which architectures are most effective.
* **Robustness testing**: The inclusion of robustness tests (though results showing fragility across all methods) is valuable for identifying limitations and areas for improvement in current approaches.
* **Unified benchmark creation**: The consolidation of four datasets into a single benchmark addresses the practical need for standardized evaluation, which could facilitate future research and fair comparisons.

**Weaknesses:**

## Weaknesses

### 1. Insufficient Literature Review and Comparative Analysis

The paper's literature review is significantly incomplete and lacks proper contextualization of the proposed work within the existing landscape of time-series text multimodal research. Specifically:

**Missing Comparative Table**: The paper does not provide a comprehensive table comparing existing datasets and benchmarks. Such a table should detail:

- What tasks each existing benchmark covers
- What data modalities and characteristics they use
- What evaluation metrics are employed
- What gaps exist that BEDTime aims to fill
- How BEDTime's contribution differs from prior work

Without this comparative analysis, it is difficult to assess the novelty and significance of the proposed benchmark. The authors should provide a detailed table (suggested format below) that clearly positions their work:

| Benchmark/Dataset      | Tasks                                    | Data Type     | Metrics                      | Key Contributions | Limitations | BEDTime's Addition |
| ---------------------- | ---------------------------------------- | ------------- | ---------------------------- | ----------------- | ----------- | ------------------ |
| [Existing Benchmark 1] | ...                                      | ...           | ...                          | ...               | ...         | ...                |
| [Existing Benchmark 2] | ...                                      | ...           | ...                          | ...               | ...         | ...                |
| BEDTime                | Recognition, Differentiation, Generation | Univariate TS | Accuracy, Generation metrics | Unified framework | ...         | ...                |

**Omitted Related Work**: The paper fails to discuss several highly relevant works that are directly related to time-series text multimodal understanding and benchmarking:

- **MTBench**: A comprehensive benchmark for multimodal time-series understanding that should be discussed and compared.
- **ITFormer**: Work on bridging time series and natural language for multi-task temporal-textual QA, which is highly relevant to the tasks proposed in this paper.
- **Time-MQA**: Multi-modal question answering for time series, which addresses similar problems and should be included in the related work discussion.

These omissions make it difficult to assess whether BEDTime offers meaningful advances over existing benchmarks or merely re-formalizes already-studied problems. The authors should provide a thorough discussion of how BEDTime relates to and improves upon these existing works.

### 2. Lack of Evidence for Downstream Task Correlation

The paper argues that descriptive capabilities are important, but provides **no empirical evidence** that performance on the proposed benchmark tasks correlates with improved performance on downstream applications. This is a critical gap for a benchmark paper, as the value of a benchmark is fundamentally tied to its ability to predict or correlate with real-world performance.

The paper should:

- Conduct experiments demonstrating correlation between BEDTime task performance and downstream task performance (e.g., forecasting, anomaly detection, classification)
- Provide theoretical or empirical justification for why these description tasks are predictive of broader time-series reasoning capabilities
- Discuss what specific downstream tasks would benefit from strong performance on recognition/differentiation/generation tasks

Without this validation, it remains unclear whether improving on BEDTime actually translates to improved real-world performance, which undermines the benchmark's utility.

### 3. Limitation to Univariate Time Series

The benchmark is exclusively evaluated on **univariate time series data**, despite the fact that multivariate time series are ubiquitous in practice and represent a fundamentally different and often more complex problem. This limitation is particularly concerning because:

- **Inter-dimensional correlations are fundamental**: The relationships between different dimensions in multivariate time series are often the most informative aspect of the data (e.g., cross-correlations in sensor networks, lead-lag relationships in financial data, spatial-temporal dependencies). A benchmark that ignores this critical characteristic may not capture the full complexity of real-world time-series understanding.
- **Architecture implications**: Models that work well on univariate data may not generalize to multivariate scenarios, where understanding relationships between dimensions is crucial. The current evaluation does not test whether the conclusions (e.g., "vision-language models perform well") hold in multivariate settings.
- **Limited practical applicability**: Many real-world applications involve multivariate time series. A benchmark that only considers univariate data has reduced practical value.

The authors should:

- Extend the benchmark to include multivariate time-series datasets
- Analyze whether the current findings (e.g., vision-language models outperforming language-only models) hold in multivariate scenarios
- Investigate whether understanding inter-dimensional relationships requires different capabilities that are not captured by the current tasks
- Discuss how the three proposed tasks (Recognition, Differentiation, Generation) should be adapted or extended for multivariate scenarios

This is not just a "future work" consideration—it is a fundamental limitation that calls into question the comprehensiveness of the benchmark as a foundational evaluation framework for time-series text multimodal understanding.

### 4. Limited Scope of Evaluation

The evaluation, while covering 13 models, could be more comprehensive:

- **Ablation studies**: More detailed analysis of why certain architectures perform better would strengthen the paper
- **Error analysis**: Systematic analysis of failure cases would help identify common patterns and areas for improvement
- **Cross-domain evaluation**: More analysis of how models generalize across the four different datasets would be valuable

**Questions:**

## Questions

1. **Literature comparison**: Can the authors provide a comprehensive table comparing existing time-series text multimodal benchmarks (including MTBench, ITFormer, Time-MQA, and others), clearly articulating what unique contribution BEDTime makes?
2. **Downstream task correlation**: Can the authors conduct experiments showing that performance on BEDTime tasks correlates with performance on downstream applications (e.g., forecasting, anomaly detection)? What is the theoretical or empirical justification for why description tasks are predictive of broader capabilities?
3. **Multivariate extension**: The paper focuses exclusively on univariate time series. Given that multivariate time series with inter-dimensional correlations are common in practice and represent a fundamentally different challenge, how do the authors plan to extend BEDTime to multivariate scenarios? Would the current conclusions about model performance change if evaluated on multivariate data?
4. **Task design rationale**: Why were these three specific tasks (Recognition, Differentiation, Generation) chosen? What is the theoretical or empirical justification that these tasks comprehensively evaluate time-series description capabilities?
5. **Dataset selection**: How were the four datasets selected? What criteria were used to ensure they are representative and sufficiently diverse?
6. **Evaluation metrics**: For the Generation task, what metrics are used to evaluate the quality of generated descriptions? How are semantic correctness, fluency, and relevance assessed?

---

> ### Author Response · Authors · 2025-11-25
> **Rebuttal Response Part-1**
>
> We thank the reviewer for their thoughtful and constructive feedback. We are encouraged by their recognition of several core strengths of our work, including the clear and rigorous task framework for assessing time-series description (Recognition, Differentiation, and Generation), our comprehensive evaluation of 13 diverse models spanning language-only, vision-language, and time-series language architectures, our inclusion of robustness tests inspired by real word perturbations to time series signals to explore model weaknesses, and the practical value of unifying four disparate datasets into a single standardized benchmark.
>
> We would also like to note that this review makes a vast number of requests, the inclusion of which would entail a paper that includes far beyond a reasonable expectation for a paper (rigorously testing a key hypothesis). We have addressed all concerns and comments below.
>
> **W1/Q1.** The paper omits a comparative tabular literature review of multimodal TS–text benchmarks (MTBench, ITFormer, Time-MQA), obscuring the novelty of BEDTime’s contribution.
>
> Thank you for this great suggestion, we plan to include the Table A below in the paper and would be happy to extend it further. We are glad this clarifies our contribution even further.
>
> |Benchmark|Tasks Evaluated|Data|Metrics|Contributions|Limitations|BEDTime's Novelty|
> |-|-|-|-|-|-|-|
> |Merrill et al., 2024[1]|Etiological reasoning;TSQA;context-aided forecasting(MCQ)|8.7k Synthetic univariate TS–text pairs; ~230k MCQs|Accuracy|Defines 3 TS reasoning task; large synthetic MCQ suite; shows LLMs struggle at zero-shot TS reasoning|Fully synthetic; no real-world signals or human descriptions; no robustness tests|Uses real TRUCE-Stock + synthetic/template data; prioritizes visually grounded descriptions; includes human evaluation and robustness tests|
> |TimeSeriesExam[2]|Pattern recognition;noise;anomaly;similarity;causality(MCQ)|Synthetic univariate TS from controlled patterns| Accuracy |Domain-agnostic TS concept exam; diagnostic breakdown | Only synthetic; MCQ-only; no open generation or real TS | Combines real and synthetic TS–text pairs; adds recognition,and open generation; uses NLI and human scoring |
> |Time-MQA[3]|Forecasting;imputation;anomaly;classification;reasoning (MCQ, T/F, open QA)| 92k QA pairs; 12 domains; real and synthetic TSw/text|Accuracy; F1| Large multi-domain TSQA dataset; frames TS tasks as QA | Many QA items machine-generated; limited VLM inclusion | Uses human, template, and synthetic sources; adds vision modality; compares LLM, VLM, and TSLM on generation tasks with NLI and human scoring |
> | ITFormer [4] | TS understanding, reasoning, decision QA for aero engines | Multivariate sensor TS (32 channels); 110k QA pairs | BLEU; ROUGE-L; accuracy; F1 | First large real TS-QA dataset; hierarchy of understanding to decision | Single domain; programmatically generated QA | Domain-agnostic benchmark over 4 datasets; generic TS patterns; 3-modality comparisons, robustness tests |
> | MTBench [5] | TS–text reasoning: forecasting; trend classification; indicator prediction; news-driven QA; TS–text correlation | Stock and weather TS aligned with news/reports | Regression errors; Accuracy |TS–text cross-modal benchmark in finance/weather | Limited domains; GPT-derived labels; causal reasoning simplified to classification | Domain-agnostic design; builds on existing labeled datasets; emphasizes TS description skills|
> | TADACap [6] | TS captioning; retrieval vs VLM captioning | Plot images from four datasets (real + synthetic) | ROUGE-L; CIDEr; SPICE; SPIDEr | Introduces domain-aware TS captioning; retrieval vs VLM comparison | Few domains; image-only; no raw TS; no multimodal evaluation | Unifies numeric 3 modalities; defines 3 tasks; has human eval and open generation |
> |CaTS-Bench[7]|TS captioning; MCQ; caption/plot matching; comparative QA| Eleven real datasets with TS, metadata, plots, LLM captions | F1; BLEU; ROUGE; METEOR|Large context-aware captioning benchmark; numeric fidelity checks; diagnostic QA|Captions mostly from one LLM (style bias); visuals limited to simple line plots|Aggregates diverse description sources; evaluates recognition, differentiation, and generation; includes robustness and multimodal comparisons|
> |BEDTime (Ours)|Recognition (T/F); differentiation (MCQ); open description;robustness tests|10,164 TS–text pairs from four datasets; human, crowd, template, and synthetic descriptions; numeric TS, plots, and generic text descriptions|Accuracy; NLI entailment; multi-criterion human scores|First unified TS description benchmark across tasks and modalities; standardized datasets;direct LLM/VLM/TSLM comparison; extensive robustness analysis; open-source|Univariate only; mostly synthetic; automated generation evaluation relies on general-purpose NLI and instruction-tuned LMs |Provides first unified TS description framework and evaluation protocol that other datasets can map onto|

---

> ### Author Response · Authors · 2025-11-25
> **Rebuttal Response Part-2**
>
> **W2/Q2.**  Correlation with downstream task performance.
>
> BEDTime is designed to test one task: time series description. As described in Section 1, this is itself an important task with multiple real use cases. While it would be nice if performance correlates with many downstream tasks, there are many other time series tasks (e.g., see the table above) and we do not believe it is necessary for performance on one benchmark to correlate with any or all others in order to be a valuable contribution. Understanding how descriptive skills relate to other capabilities is therefore just an interesting future direction. To prevent misinterpretation, we will revise the introduction to ensure such downstream performance is unmotivated and clarified in the Limitations section acknowledging that, while downstream correlations may be valuable to explore, BEDTime is intentionally scoped as a benchmark for the descriptive task only.
>
> **W3/Q3.**	The benchmark only covers univariate time series, missing the complexity of real-word multivariate time series data and raising questions about whether the current results would hold in more realistic settings.
>
> While multivariate time series are of course important, they are clearly not “more realistic” than univariate time series, which are extremely prevalent in many many domains. This is why they have been extensively studied for decades. Therefore, we must respectfully but firmly disagree with this being a limitation of our work. The specific listed requests are more than what would fill an entire new paper. So while we are excited about multivariate extensions, too, no appropriately-labeled data exist, so studying their description will require large data annotation efforts that must be reserved for future, dedicated efforts, which we believe will turn up many interesting, new findings on their own.

---

> ### Author Response · Authors · 2025-11-25
> **Rebuttal Response Part-3**
>
> **W4.** Lacking ablations to explain model failure cases, systematic error analysis, and examination of how models generalize across domains.
>
> We appreciate the reviewer’s suggestions and respectfully clarify several points about our scope and scale:
>
> - BEDTime evaluates 13 models of varying parameter sizes across 3 modalities (6 LLMs, 4 VLMs, 2 TSLMs, and a BPE-dependent prompting variant for GPT (LLMTime)), across 4 datasets spanning synthetic, semi-synthetic, and real-world domains, 3 tasks (description recognition, description differentiation, open generation), 4 distractor-selection strategies (DTW Distance, Euclidean Distance, Sentence-BERT Embeddings with Cosine Similarity and LCSS Similarity). We also have both automatic and human evaluation for open-generation. We have also conducted 5 real world perturbation experiments to explore further failure cases(sequence length (interpolation and scaled interpolation), missingness, amplitude scaling, Gaussian noise, CoT prompting).
>
> - For ablations, it is unclear how to ablate the big, pre-trained models we are intentionally evaluating. Some of them are also proprietary. . For most models (GPT-4o, Gemini-2.0, etc.), internal architectural components cannot be ablated. For open-source models (e.g., Llama, Qwen and Phi), modifying visual encoders, tokenization schemes, or temporal modules would require retraining these models, which is clearly outside the scope of a benchmark paper. If the reviewer has specific ablations in mind, we welcome concrete suggestions and would be happy to incorporate them in future work.
>
> - For cross-domain generalization, we actually already measure this in multiple ways. Cross-domain generalization is directly measured by:
> Using four datasets with very different visual, statistical, and semantic structures (TRUCE-Stock, TRUCE-Synthetic, SUSHI, TaxoSynth). (Tables 5-12)
> Reporting results separately per dataset and comparing intra-modal rank consistency across datasets. (Tables- 2,3 and Figure-4)
> We also already provide a substantial error analysis throughout the paper. It being a benchmark paper, this is actually a large portion of our analysis. For example, we include:
> - Failure-case characterization for TSLMs (ChatTime refusal rates in Appendix F)
> Sensitivity patterns under sequence length (interpolation and scaled interpolation), missingness, amplitude scaling, Gaussian noise, CoT prompting. (Figure 2,3)
>
> So ultimately, while we agree that many additional analyses could be explored in future work, we think these extra experiments are either completed already or unnecessary for our contribution. We will clarify the lack of any such ablations in the Limitations sections.
>
>
> **Q4.** Explaining the rationale for selecting the three tasks and how they collectively capture time-series description ability would strengthen the overall task design.
>
> We chose Recognition, Differentiation, and Generation because, taken together, they reflect the most common and practically useful modes for evaluating generative models in multimodal and language-based settings and dominate prior work on time-series language benchmarks. As summarized in Table A in **W1/Q1.**, existing benchmarks, irrespective of their evaluation goals like: synthetic reasoning suites, diagnostic MCQ exams, or domain-specific QA datasets, ultimately evaluate models through some combination of verification (true/false or equivalence judgments), selection (multiple-choice or ranking among candidate descriptions), and open-ended generation. Our design follows this established structure: Recognition and Differentiation provide simple, scalable, automatic, accuracy-based evaluations that serve as widely used proxies for assessing whether a model can verify or discriminate descriptions, even if they do not fully capture real-world usage. Generation, by contrast, represents the realistic setting in which a model must produce an open-form description of a time series, but its evaluation is inherently difficult: LLM-as-judge and NLI-based methods offer automated, though sometimes biased, signals, and human evaluation remains the gold standard despite its cost. Our goal is not to claim that these are the only tasks, but rather that they provide an exhaustive, reasonable, and well-established set of foundations for assessing descriptive ability. This rationale aligns with the BEDTime's motivation, namely, benchmarking language models' abilities to describe the visual properties of time series, as discussed in Figure 1 and Section 3 of the manuscript.

---

> ### Author Response · Authors · 2025-11-25
> **Rebuttal Response Part-4**
>
> **Q5.**  Justification is needed for how the four datasets were selected to be representative and diverse.
>
> We clarify that we selected the four datasets in BEDTime by relying exclusively on previously peer-reviewed or publicly released datasets that contained time series paired with visual descriptions of how the values change over time, and with established, rigorously documented curation procedures. These four datasets were the best fits as far as we know after an extensive literature review: TRUCE-Stock and TRUCE-Synthetic , SUSHI , and TaxoSynth. TRUCE-Stock uses MTurk crowd descriptions collected under strict annotator filters, concealment of stock identifiers to prevent bias, short caption limits to isolate single patterns, and multi-annotator redundancy with manual quality control. TRUCE-Synthetic employs controlled functional generators with light noise to ensure clean ground-truth alignment. SUSHI includes synthetic signals drawn from predefined functional classes paired with template-based descriptions that were manually refined and visually validated for accuracy. TaxoSynth is generated from a rigorous behavioral taxonomy with parameterized functional templates and dataset-level validation checks to ensure each series faithfully expresses the intended pattern class.
>
> To further explore whether these four datasets are representative of model behavior, we have added a new real-world, clinically-grounded dataset containing NICU heart-rate time series signal data, patient metadata, and validated clinical descriptions [8]. We describe the dataset and experiments below, which will also be added to the paper.
>
> The added NICU dataset includes 36,679 public real heart-rate recordings from infants treated in a NICU from 2012–2016. Each 10-minute (300-step, 2-second sampled) sequence is paired with:
>
> - Natural language clinical descriptions derived from clinical bradycardia events and variability annotations that were clinically validated for semantic diversity and correctness.
>
> - Patient metadata, that includes demographics, delivery mode, Apgar scores, comorbidities, gestational age, birth weight, maternal age, race, etc.
>
> We evaluate whether models can identify real NICU heart-rate signals’ clinical descriptions through two tasks:
>
> - Description Recognition (True/False): Models judge whether a candidate description accurately reflects the heart rate time series signal. Here:
> The distractors come are chosen as the most different signal based on via DTW-distance. Distractors selection is constrained as to make sure that they always come from patients with opposite clinical outcomes.
>
> - Description Differentiation (Multiple Choice): Models select the correct description among four real clinical descriptions (1 correct and 3 distractors chosen in the same constraints as above).
>
> Both tasks are repeated with and without patient metadata, requiring models to integrate physiological patterns with etiological factors rather than relying only on visual cues. The table below shows the results for these experiments, which all align with our existing findings on the synthetic data. Bolstering our previous BEDTime results, VLMs lead across settings, with GPT-4o-Vision at the top. GPT-4o takes the lead amongst LLMs and benefits significantly with BPE-based LLMTime sequence inputs. As seen previously, TSLMs beat their base LLM baselines with ChatTS performing better than all LLMs of comparable size across all 4 settings. The introduction of Patient Metadata shows an interesting pattern: While there is an overall increase in model performance when patient metadata is introduced. Larger models(>14B) show greater improvement over lower parameter models (<7B) in this setting across both tasks.
>
> |Models|Recognition(TS)|Recognition(TS+Metadata)|Avg. Recognition Rank| Differentiation(TS)|Differentiation(TS+Metadata)|Avg. Differentiation Rank|Overall Rank|
> |-|-|-|-|-|-|-|-|
> |GPT-4o|0.567|0.640|2|0.944|0.971|2|2|
> |GPT-4o+LT|0.569|0.645|1|0.948|0.979|1|1|
> |Gemini-2.0|0.557|0.638|3|0.938|0.969|3|3|
> |Llama-3.1-8B|0.458|0.447|7|0.689|0.678|6.5|6.75|
> |Qwen2.5-14B|0.540|0.612|4|0.814|0.852|4|4|
> |Qwen2.5-7B|0.530|0.520|5|0.782|0.698|5|5|
> |Phi-3.5-mini|0.506|0.518|6|0.765|0.667|6.5|6.25|
> |GPT-4o-Vision|0.604|0.667|1|0.957|0.983|1|1|
> |Gemini-2.0-Vision|0.597|0.613|2|0.943|0.976|2|2|
> |Qwen2.5-VL-7B|0.544|0.553|3|0.793|0.721|3|3|
> |Phi-3.5-Vision|0.529|0.529|4|0.771|0.714|4|4|
> |ChatTime-7B|0.543|0.562|2|0.727|0.711|2|2|
> |ChatTS-14B|0.545|0.587|1|0.821|0.776|1|1|
>
> These experiments show that while adding and exploring new data is valuable, it does not alter our conclusions and BEDTime remains to be representative of model behavior on external and out-of-distribution real-world data. We will incorporate this NICU study as an additional real-world case of descriptive and etiological time-series reasoning.

---

> ### Author Response · Authors · 2025-11-25
> **Rebuttal Response Part-5**
>
> **Q6.** Clarification is needed on how Generation outputs are evaluated, specifically, which metrics assess semantic correctness, fluency, and relevance.
>
> Our evaluation of Generation outputs relies on a mix of both, Natural Language Inference Entailment and human annotation, where human evaluation (based on six criteria from prior literature) serves as the primary measure. Semantic correctness is captured through three criteria: **Pattern Identification**(the “Patterns” column in Table 3), which checks whether the description correctly identifies key features such as trends, spikes, or cycles; **Temporal Localization**(the “Position” column in Table 3), which evaluates whether those features are placed at the correct points in time; and **Noise Filtering**(the “Noise” column in Table 3), which ensures the model ignores insignificant fluctuations rather than misinterpreting them. Together, these criteria assess whether the generated text accurately reflects the underlying time-series signal. Fluency is measured through **Coherence**(the “Valid” column in Table 3), which ensures the description is logically connected and meaningful, and **Linguistic Quality**(the “Concise” column in Table 3), which evaluates whether the language is clear, concise, and well-formed. Finally, relevance is assessed through **Abstraction** (the “Step-by-Step” column in Table 3), which verifies that the model focuses on meaningful high-level structure rather than unnecessary point-by-point repetition. Inter-annotator agreement for these criteria based evaluations was high, with Cohen’s Kappa and Krippendorff’s Alpha exceeding 0.8 for all criteria.
>
> DeBERTa NLI is used as an auxiliary signal. To address numeric mismatch, we compute NLI in two modes: (1) with numeric descriptions (“w/#”) and (2) with numeric descriptions removed (“w/o #”) to prevent absolute numeric values from influencing entailment. All three evaluation modes—w/# NLI, w/o # NLI, and human ratings—produce the exact same ranking on SUSHI, shown below:
>
> |Model|Average w/# NLI Rank|Average w/o # NLI Rank|Average Human Rater Rank|Overall Rank|
> |-|-|-|-|-|
> |GPT-4o-Vision|1.00|1.00|1.00|1.00|
> |GPT-4o|2.33|2.33|2.33|2.33|
> |ChatTS-14B|2.67|2.67|2.17|2.50|
>
> While raw DeBERTa entailment is extremely sensitive to lexical and syntactic variation. To further address this domain mismatch, we additionally evaluate generation using a GPT-5 as an LLM judge, which is known to perform far better on semantic alignment and paraphrase recognition than DeBERTa. In this setup, for each pair consisting of the ground-truth description and the model-generated description, we explicitly prompt GPT-5 to decide whether the generation entails the ground truth, whether the ground truth entails the generation, and if both conditions hold simultaneously we treat it as bi-directional entailment. This yields the following entailment-style scores, which again preserve the exact same model ranking:
>
> |Model|w/# Bi-dir %	|w/# Gen→GT %|w/# GT→Gen %|w/# Average Rank|
> |-|-|-|-|-|
> |GPT-4o-Vision |47.64|71.43|51.86|1|
> |GPT-4o|37.93|58.14|43.43|2|
> |ChatTS-14B|20.93|46.71|34.93|3|
>
> REFERENCE:
>
> [1] Merrill et al., 2024. "Language Models Still Struggle to Zero-shot Reason about Time Series".
>
> [2] Cai et al., 2024. "TimeSeriesExam: A Time Series Understanding Exam".
>
> [3] Kong et al., 2025. "Time-MQA: Time Series Multi-Task Question Answering with Context Enhancement".
>
> [4] Wang et al., 2025. "ITFormer: Bridging Time Series and Natural Language for Multi-Modal QA with Large-Scale Multitask Dataset".
>
> [5] Chen et al., 2025. "MTBench: A Multimodal Time Series Benchmark for Temporal Reasoning and Question Answering".
>
> [6] Fons et al., 2025. "TADACap: Time-series Adaptive Domain-Aware Captioning".
>
> [7] Zhou et al., 2025. "CaTS-Bench: Can Language Models Describe Numeric Time Series?".
>
> [8] Qiu et al., 2025. "Instruction-based Time Series Editing".

---

### Official Review · Reviewer_HnSu · 2025-11-01

**Soundness:** 3
**Presentation:** 3
**Contribution:** 3
**Rating:** 4
**Confidence:** 3

**Summary:**

The paper introduces a benchmark that evaluates a model's capability to recognize, differentiate, and generate language descriptions of time series. The benchmark aggregates four datasets and evaluates 13 models spanning three paradigms, including language models, vision language models, and time series language models. Results show that overall vision language models outperform the other two types.

**Strengths:**

1. The paper presents the first benchmark explicitly focused on language-based understanding of time-series characteristics, enabling apples-to-apples comparisons across modalities.

2. The paper compares a large set of models spanning three types of popular paradigms. It also conducts a series of robustness studies given e.g., missing values, varying scales and noise levels to provide insights on current methods.

**Weaknesses:**

1. The benchmark contains mostly synthetic datasets and statistical properties. It would be helpful to generalize to more real-world datasets, and incorporates semantic grounding, for example describing the potential domains that the time series come from (etiological reasoning).

2. The study primarily shows that existing models fall short, but does not propose methods to address these gaps. In addition, the evaluation focuses on zero-shot performance and does not consider fine-tuned models.

3. The ability of describing a time series does not necessarily translate to other tasks. For instance, visual encodings can capture global shape but lose numerical precision, so VLMs that excel at descriptions might still underperform on forecasting or numerically sensitive tasks.

**Questions:**

Minor: Better to show the overall rankings in Table 2 as well.

---

> ### Author Response · Authors · 2025-11-25
> **Rebuttal Response Part-1**
>
> We thank the reviewer for their thoughtful and constructive feedback. We are encouraged they recognize our benchmark’s novelty as the first to explicitly evaluate language models as time-series descriptors, the breadth of our side-by-side comparison across three input modalities, and the thoroughness of our robustness studies under realistic perturbation settings. We address the reviewer’s concerns and comments below:
>
> **W1.** BEDTime relies mostly on synthetic data with statistical properties; expanding to real-world data and adding semantic grounding would be helpful.
>
> We thank the reviewer for this thoughtful comment. We agree that our data are mostly synthetic (though one already includes hand-written descriptions). To further address this feedback, we have added a new real-world, clinically-grounded dataset containing NICU heart-rate time series signal data, patient metadata, and validated clinical descriptions [1]. We describe the dataset and experiments below, which will also be added to the paper.
> The added NICU dataset includes 36,679 public real heart-rate recordings from infants treated in a NICU from 2012–2016. Each 10-minute (300-step, 2-second sampled) sequence is paired with:
>
> - Natural language clinical descriptions derived from clinical bradycardia events and variability annotations that were clinically validated for semantic diversity and correctness.
>
> - Patient metadata, that includes demographics, delivery mode, Apgar scores, comorbidities, gestational age, birth weight, maternal age, race, etc.
>
> We evaluate whether models can identify real NICU heart-rate signals’ clinical descriptions through two tasks:
>
> - Description Recognition (True/False): Models judge whether a candidate description accurately reflects the heart rate time series signal. Here:
> The distractors come are chosen as the most different signal based on via DTW-distance. Distractors selection is constrained as to make sure that they always come from patients with opposite clinical outcomes.
>
> - Description Differentiation (Multiple Choice): Models select the correct description among four real clinical descriptions (1 correct and 3 distractors chosen in the same constraints as above).
>
> Both tasks are repeated with and without patient metadata, requiring models to integrate physiological patterns with etiological factors rather than relying only on visual cues. The table below shows the accuracy results for these experiments, which all align with our existing findings on the synthetic data. Bolstering our previous BEDTime results, VLMs lead across settings, with GPT-4o-Vision at the top. GPT-4o takes the lead amongst LLMs and benefits significantly with BPE-based LLMTime sequence inputs. As seen previously, TSLMs beat their base LLM baselines with ChatTS performing better than all LLMs of comparable size across all 4 settings. The introduction of Patient Metadata shows an interesting pattern: While there is an overall increase in model performance when patient metadata is introduced. Larger models(>14B) show greater improvement over lower parameter models (<7B) in this setting across both tasks.
>
> |Models|Recognition(TS)|Recognition(TS+Metadata)|Avg. Recognition Rank| Differentiation(TS)|Differentiation(TS+Metadata)|Avg. Differentiation Rank|Overall Rank|
> |-|-|-|-|-|-|-|-|
> |GPT-4o|0.567|0.640|2|0.944|0.971|2|2|
> |GPT-4o+LT|0.569|0.645|1|0.948|0.979|1|1|
> |Gemini-2.0|0.557|0.638|3|0.938|0.969|3|3|
> |Llama-3.1-8B|0.458|0.447|7|0.689|0.678|6.5|6.75|
> |Qwen2.5-14B|0.540|0.612|4|0.814|0.852|4|4|
> |Qwen2.5-7B|0.530|0.520|5|0.782|0.698|5|5|
> |Phi-3.5-mini|0.506|0.518|6|0.765|0.667|6.5|6.25|
> |GPT-4o-Vision|0.604|0.667|1|0.957|0.983|1|1|
> |Gemini-2.0-Vision|0.597|0.613|2|0.943|0.976|2|2|
> |Qwen2.5-VL-7B|0.544|0.553|3|0.793|0.721|3|3|
> |Phi-3.5-Vision|0.529|0.529|4|0.771|0.714|4|4|
> |ChatTime-7B|0.543|0.562|2|0.727|0.711|2|2|
> |ChatTS-14B|0.545|0.587|1|0.821|0.776|1|1|
>
> From this we can conclude that while addition of new datasets is beneficial to BEDTime, the benchmark with its 4 datasets is representative of model performance on external/out-of-distribution data and our conclusions made in the paper holds. We will update our main manuscript to introduce this as a case-study on out of distribution data with etiological reasoning.

---

> ### Author Response · Authors · 2025-11-25
> **Rebuttal Response Part-2**
>
> **W2.** BEDTime highlights current model shortcomings but does not propose methods to address them.
>
> Our submission is intentionally listed as a “datasets and benchmarks” primary subject area, which is explicitly sought by ICLR in the call for papers.So BEDTime is intentionally scoped as a benchmarking contribution rather than a method development contribution. We would also like to clarify that our experiments do include fine-tuned baselines, as both ChatTS and ChatTime explicitly report being fine-tuned for time-series-and-language tasks. ChatTS states that it performs “understanding and reasoning about time series attributes” via synthetic time-series–text instruction tuning, and ChatTime describes itself as an “out-of-the-box multimodal time series foundation model” trained through continuous pre-training and instruction fine-tuning to support time-series-to-text generation and question answering. While untrained on our data, we are confident they should be expected to perform reasonably well on our data based on those authors’ claims. To ensure this scope of BEDTime is clear to future readers, we will explicitly revise the introduction to state BEDTime is a benchmark rather than a method, add a note in the Limitations stating that method development lies outside our present scope, and clarify that fine-tuning models on BEDTime is an interesting direction for future research, but the current work is intended strictly as an evaluation benchmark.
>
> **W3.** Evaluating transfer to downstream tasks.
>
> We thank the reviewer for this observation, and we agree that strong descriptive ability does not automatically imply good performance on any downstream. Moreover, such transfer is not our focus as BEDTime is not intended to directly predict downstream task performance, and we do not claim that descriptive ability is necessary for such numerically-sensitive tasks. Instead, BEDTime focuses on a stand-alone description task, with independent motivation and real world applications, as described in the introduction of our main manuscript. Additionally, models such as ChatTS and ChatTime are described as general-purpose “time series foundation models,” making it reasonable to evaluate whether they can reliably handle basic descriptive tasks alongside VLMs. To avoid any overinterpretation, we will clarify in the introduction that BEDTime does not assume any relationship to downstream tasks.
>
> **Q1.** Add overall model rankings in Table 2.
>
> We agree that including both overall and modality-stratified rankings strengthens the analysis, and have added them below and will integrate them into the manuscript. These rankings give way to the most obvious trend, that models having visual time-series inputs is the dominant factor shaping performance outweighing model size, architectural differences, and even time series specific pre-training for the task of time series description.
>
> |Model|Overall Recognition Rank|Overall Differentiation Rank|Overall Rank|
> |-|-|-|-|
> |GPT-4o|5|5.25|5.125|
> |GPT-4o+LT|2.75|3.75|3.25|
> |Gemini-2.0|6.25|6.75|6.5|
> |Llama-3.1-8B|9.75|11.25|10.5|
> |Qwen2.5-14B|7.25|7.25|7.25|
> |Qwen2.5-7B|7.5|9.5|8.5|
> |Phi-3.5-mini|9|9.75|9.375|
> |GPT-4o-Vision|1|1|1|
> |Gemini-2.0-Vision|3|2.75|2.875|
> |Qwen2.5-VL-7B|6|3|4.5|
> |Phi-3.5-mini-vision|4.5|4.5|4.5|
> |ChatTime-7B|11.25|11.75|11.5|
> |ChatTS-14B|6.75|7.75|7.25|
>
> We have also augmented Table 3 to include overall model rankings, computed by averaging each model’s rank across all automated metrics, both with and without numerical descriptions (bi-directional entailment, Gen→GT, and GT→Gen), as well as all six human-evaluation criteria. The updated table appears in the revised manuscript. These aggregated rankings reinforce the main paper’s conclusions: GPT-4o-Vision remains the top-ranked model overall, followed by GPT-4o, with ChatTS performing closely behind.
>
> |Model|w/# Bi-dir %|w/# Gen→GT %|w/# GT→Gen %|w/# NLI Rank|w/o # Bi-dir %|w/o # Gen→GT %|w/o # GT→Gen %|w/o # NLI Rank|Valid|Patterns|Position|Noise|Step-by-Step|Concise|Human Rater Rank|Overall Rank
> |-|-|-|-|-|-|-|-|-|-|-|-|-|-|-|-|-|
> |GPT-4o-Vision|**14.41**|**47.65**|**15.29**|**1.00**|**15.76**|**49.84**|**16.72**|**1.00**|**1.00**|**0.891**|**0.751**|**0.815**|**0.853**|**0.851**|**1.00**|**1.00**|
> |GPT-4o|2.94|9.71|6.18|**2.33**|3.22|10.61|6.43|**2.33**|1.00|0.361|0.234|0.624|0.681|0.715|**2.33**|**2.33**|
> |ChatTS-14B|2.65|23.53|2.65|**2.67**|2.89|25.40|2.89|**2.67**|1.00|0.738|0.540|0.657|0.613|0.620|**2.17**|**2.50**|
>
> REFERENCE:
>
> [1] Qiu et al., 2025. "Instruction-based Time Series Editing".

---

### Author Response · Authors · 2025-12-03
**Final Author Statement**

Dear AC,

We thank you for the time and care put into evaluating our work. We are pleased by the positive reception of our contributions to the literature.  Across all four reviewers, several consistent strengths were highlighted, as follows:

BEDTime is “the first benchmark explicitly focused on language-based understanding of time-series characteristics, enabling apples-to-apples comparisons across modalities.” (Reviewer HnSu) that “addresses an important evaluation gap” (Reviewer iBNc), providing “well-defined task framework… Recognition, Differentiation, Generation clearly capture different reasoning types” (Reviewer KZgR) with“comprehensive evaluation of 13 diverse models … revealing valuable patterns.” (HnSu, KZgR). BedTime  includes “Extensive robustness studies (missingness, varying scales, noise) provide insights.” (HnSu). BedTime has “Exceptionally thorough experimental coverage … easy to trace claims to specific results.” (iBNc) that brings out “surprising and actionable finding that VLMs consistently outperform specialized TSLMs.” (iBNc). Notably both Reviewer za99 and iBNc were explicitly noted that our responses successfully answered their concerns and our additional analyses further strengthened BEDTime.

To directly address reviewer concerns, we added several focused analyses during rebuttal:

- **Real-world NICU dataset added** containing 36,679 clinical heart-rate sequences with validated descriptions and rich metadata. The same modality ranking holds, confirming that the original four datasets are representative and that our conclusions generalize to real, out-of-distribution settings.

- **Expanded related-work comparison** through a structured table contrasting BEDTime with MTBench, Time-MQA, ITFormer, TimeSeriesExam, TADACap, CaTS-Bench, and others, making clear where prior benchmarks fall short for TS–text evaluation and how BEDTime fills those gaps.

- **Pixelation robustness study added** using controlled downsampling (×2–×32) on SUSHI, showing monotonic accuracy degradation for all VLMs and demonstrating that BEDTime meaningfully probes visual robustness and that VLM performance relies on genuine visual structure.

- **Generation-evaluation validation performed** using human ratings, NLI both with and without numeric tokens, and LLM-judge entailment, all of which produce the same ranking (GPT-4o-Vision > GPT-4o > ChatTS), confirming that generation results are consistent across metrics.

- **Clarified TSLM landscape** by showing that most suggested TSLMs cannot perform TS–text description tasks (lack text generation, lack TS–text semantic alignment, or have no released weights). We evaluate all publicly viable models that do support these tasks (ChatTS, ChatTime), including refusal-rate analysis.

- **Clarifications on sampling and task design provided** by explaining that negatives always come from real descriptions (never generated), are sampled using multiple distance metrics with class constraints, and are fully deterministic. BEDTime intentionally targets global visual descriptions, and strong performance drops under perturbations confirm the tasks are non-trivial.

In summary, we believe our rebuttal directly addressed every major concern with new analyses and real-world validation, leading two reviewers to raise their scores making BEDTime a clear, well-supported infrastructure contribution for standardized, multimodal time-series evaluation.

---

### Meta-Review · Area_Chair_zm2s · 2026-01-02

**Summary:**

Reviewers generally agree that BEDTime targets an important gap and is carefully executed, with broad experiments and clear takeaways. However, several concerns prevent it from clearly meeting the ICLR bar. These include doubts about benchmark scope and representativeness (heavy reliance on synthetic and univariate data), fairness and interpretation of cross-modality comparisons, and whether some headline claims (e.g., TSLM underperformance) are sufficiently supported. While the benchmark is useful, some reviewers felt its conceptual positioning, evaluation design, and diagnostic depth remain incomplete for a flagship dataset paper.

**Reviewer Concerns:**

The rebuttal was strong and addressed many concrete issues: it substantially improved the literature comparison, clarified task design and negative sampling, added robustness studies, expanded generation evaluation, and introduced a real-world NICU dataset. These efforts resolved several medium-level concerns and led some reviewers to raise scores. That said, core disagreements remain: the reliance on mostly synthetic and univariate data, limited coverage of functional TSLMs, and lingering questions about fairness and generalizability across modalities. These unresolved points ultimately weigh against acceptance.

**Reviewer Scores:**

Reviewer HnSu: Likely unchanged

Reviewer KZgR: Possibly a slight increase, given the expanded literature table and clarifications, but still below accept due to scope limitations.

Reviewer za99: Already increased score after rebuttal

Reviewer iBNc: Likely increased slightly

---

### Decision · Program_Chairs · 2026-01-26

Reject